

# Potential of using $CO_2$ observations over India in regional carbon budget estimation by improving the modelling system

Vishnu Thilakan[1,2], Dhanyalekshmi Pillai[1,2], Jithin Sukumaran[1,2], Christoph Gerbig[3], Haseeb Hakkim[4], Vinayak Sinha[4], Yukio Terao[5], Manish Naja[6], and Monish Vijay Deshpande[1,2]

[1]Indian Institute of Science Education and Research Bhopal (IISERB), India.
[2]Max Planck Partner Group (IISERB), Max Planck Society, Munich, Germany
[3]Max Planck Institute for Biogeochemistry, Jena, Germany
[4]Indian Institute of Science Education and Research Mohali (IISERM), India.
[5]Earth System Division, National Institute for Environmental Studies (NIES), Japan
[6]Atmospheric Science Division, Aryabhatta Research Institute of Observational Sciences, Nainital, India

**Correspondence:** Dhanyalekshmi Pillai (dhanya@iiserb.ac.in)

**Abstract.** Devising effective national-level climate action plans needs a more detailed understanding of the regional distribution of sources and sinks of greenhouse gases. Due to insufficient observations and modelling capabilities, India's current carbon source-sink estimates are uncertain. This study uses a high-resolution transport model to examine the potential of $CO_2$ observations over India for inverse estimation of regional carbon fluxes. We make use of four different sites in India that vary

in measurement technique, frequency and spatial representation. These observations exhibit substantial seasonal (7.5 to 9.2 ppm) and intra-seasonal (2 to 12 ppm) variability. Our modelling approach, a high-resolution Weather Research and Forecasting Model combined with Stochastic Time Inverted Lagrangian Transport (WRF-STILT) model, performs better in simulating seasonal ($R^2$ = 0.50 to 0.96) and diurnal ($R^2$ = 0.96) variability of observed $CO_2$ than the current generation global models analysed in the study. Representation of local flux variability like biomass burning in the model needs further refinement,

depending on the site location. During the agricultural season, crop biospheric uptake in the Indo-Gangetic Plain region significantly modulates the $CO_2$ variability in the northern Indian stations. Depending on the region and time of the year, the anthropogenic and biospheric emission components contribute differently to $CO_2$ variability. The choice of emission inventory in the modelling framework alone leads to significant biases in simulations (5 to 10 ppm), endorsing the need for accounting emission fluxes, especially for non-background sites. By implementing a high-resolution model, our results emphasise that

observations from Indian sites can be useful in deducing carbon flux information at regional (Nainital) and sub-urban to urban (Mohali, Shadnagar, Nagpur) scales. On accounting for observed variability, the global carbon data assimilation system can thus benefit from the measurements from the Indian subcontinent.

## 1   Introduction

The global terrestrial ecosystem acts as a significant carbon sink. A decrease in sink capacities accelerates global warming

as a consequence of the increased atmospheric emission fraction (airborne fraction). How the terrestrial carbon sink capacity responds to the rate of atmospheric greenhouse increase remains uncertain, implying large uncertainties in future climate



predictions. Further, significant uncertainties exist in our estimations of the magnitude and spatial distribution of carbon fluxes between land, atmosphere, and oceans (Friedlingstein et al., 2022). These estimates are particularly critical to devise effective mitigation plans for climate change. The carbon budget estimation system must sufficiently represent the complex exchange

processes operating at different spatial and temporal scales to address the above key shortcoming.

India needs an accurate estimation of its carbon sources and sinks to achieve its Nationally Determined Contribution (NDC) goals (https://unfccc.int, last access: 25 March 2023) through emission reduction. The bottom-up approach is widely used to estimate carbon fluxes based on our prior knowledge of the processes determining the fluxes, such as vegetation, land types, and fossil fuel usage statistics. However, these estimates often suffer from large errors due to various factors, including the

reliability of statistical reports, the accuracy of flux estimation approaches, and desired spatiotemporal resolution. An inverse modelling framework (top-down approach, Enting (2002)) encompassing atmospheric transport models and observations of atmospheric carbon concentrations have the potential to improve bottom-up based estimates of the source-sink distribution of carbon globally (e.g., Rödenbeck et al., 2003; Peters et al., 2007; Inness et al., 2019) and on regional scales (e.g., Gerbig et al., 2003; Lauvaux et al., 2009; Broquet et al., 2013; Gerbig et al., 2009; Pillai et al., 2016). There have been a few recent

attempts to estimate the carbon fluxes over the South Asian region using inverse modelling techniques (e.g., Patra et al., 2013; Thompson et al., 2016; Ganesan et al., 2017; Gahlot et al., 2017). It is recognised that a major source of uncertainty in inverse estimations is the lack of observational data with sufficient temporal and spatial coverage (Patra et al., 2013; Thompson et al., 2014).

In-situ observations are essential for the tropics because satellite observations cannot always detect surface variations in

addition to data gaps due to clouds and moist convection. India has recently added a few greenhouse gases (GHG) monitoring stations to its observational network (e.g., Tiwari et al., 2014; Lin et al., 2015; Mahesh et al., 2015; Nomura et al., 2021). High-frequency observations with diurnal and synoptic variations provide information on the regional sources and sinks for atmospheric $CO_2$, which are influenced by mesoscale atmospheric transport (Law et al., 2002; Gerbig et al., 2003; Geels et al., 2004; Lin et al., 2004; Lauvaux et al., 2008). $CO_2$ anomalies generated remotely can also affect these observation sites

through horizontal advection. Law et al. (2002) has suggested that the use of high-frequency observation can aid in reducing the uncertainty in inverse estimates, similar to using a larger observation network with low-frequency observations. However, measurements obtained from an observation site close to a variable source or meteorologically complex areas are difficult to represent in the transport models used for inversions. Owing to these constraints, none of the current generation global carbon assimilation systems utilises $CO_2$ observations from the Indian region to optimize their models. To utilise the potential of

these observations through inverse modelling, we need to improve our understanding of the processes driving high-frequency variability in these measurements (Geels et al., 2004). That is, sufficient improvement in modelling capabilities is required over the Indian region.

The skill of the model is determined by how well it can simulate the variability in atmospheric $CO_2$ concentration associated with transport (e.g., advection and vertical mixing) and flux distribution (e.g., anthropogenic emissions and biospheric fluxes).

Most current generation carbon flux estimations over India are derived from global carbon estimates, which utilise coarse-resolution transport models (e.g., Rödenbeck et al., 2003; Peters et al., 2007; Inness et al., 2019) for their simulations. However,



atmospheric $CO_2$ exhibits strong spatiotemporal variations such that the transport models need a horizontal resolution higher than 30 km to represent the variability (Gerbig et al., 2003). Similarly, local and large-scale convections play a major role in distributing tracer concentrations (Gerbig et al., 2003) vertically, which is difficult to simulate in tropical regions (Thompson et al., 2014). Fine-scale features are better resolved when the horizontal resolution of transport models is increased (Geels et al., 2007; Tolk et al., 2008; Agustí-Panareda et al., 2019). Considerable representation errors exist when we use coarse-resolution transport models for inverse optimization over India, and the representation error tends to decrease when we increase the horizontal resolution (Thilakan et al., 2022). The seasonally reversing monsoon circulation pattern and complex topography complicate regional atmospheric transport, influencing the vegetation patterns and agricultural practices over the region. Hence, an adequate representation of the atmospheric $CO_2$ distribution over India relies on a modelling system that can operate at high spatial and temporal resolution.

Here, we examine the capability of a high-resolution modelling framework based on the Lagrangian particle dispersion model (LPDM) to simulate $CO_2$ variability over four different observation sites in India. We follow the receptor-oriented framework described in Gerbig et al. (2003) using an LPDM called the Stochastic Time-Inverted Lagrangian Transport (STILT) model (Lin et al., 2003). A receptor-oriented analysis framework was designed to quantify the sensitivity of the atmospheric $CO_2$ concentrations (influence functions) at measurement locations (receptors) to the surface fluxes in the upwind regions or boundary conditions and thereby interpret the atmospheric signatures of the surface processes. These influence functions (footprints) can be considered equivalent to the adjoint of the Eulerian transport model. This STILT modelling framework utilises meteorology from an Eulerian transport model, surface fluxes from biospheric models or inventories, and boundary conditions from global reanalysis products to simulate the atmospheric $CO_2$ concentration at receptor locations. The boundary conditions are indented to provide the background and influence of remote fluxes on the observations. In this study, the Weather Research and Forecasting model (WRF) is used to simulate meteorology at a horizontal resolution of 10 km × 10 km and a temporal resolution of one hour. Using STILT has the advantage of simulating $CO_2$ variability down to spatial scales that are slightly smaller than the grid size of the meteorological fields used (Lin et al., 2003; Gerbig et al., 2003). Because it employs a backward time simulation strategy, it is more computationally cost-effective than an alternative forward time simulation (Lin et al., 2003), at least for data-sparse situations with only a few observational sites. The $CO_2$ observations used in this study were taken from the near-surface using different measurement techniques at different frequencies. We assess the usability of these measurements in the inverse framework when utilising the high-resolution (e.g., WRF-STILT) modelling system to optimise carbon fluxes. We quantify the model uncertainties and compare them with those of existing models.

This paper is organised as follows: Sect. 2 briefly describes the modelling framework employed in this study. In Sect 3, we provide the details of $CO_2$ measurements and global reanalysis data used in this study. Sect. 4 deals with the methods used for assessing the model skill in capturing observed variability. Sect. 5 presents the observed $CO_2$ variability across India, investigating how well STILT and global models could capture these variations. In Sect. 6, we further discuss the potential of using these observations in the future inverse modelling system, taking into account the current limitations of our modelling system. The conclusions are presented in Sect. 7.



## 2 Modelling framework

We simulated $CO_2$ concentrations at the measurement locations using the WRF-STILT modelling framework. A detailed description of the WRF-STILT system can be obtained from Nehrkorn et al. (2010). The STILT is a widely used LPDM to determine the influence of surface emissions at a receptor location by simulating the transport in the near field (i.e. the surface that planetary boundary layer (PBL) air has come into contact with before arriving at the measurement location (e.g., Lin et al., 2003; Gerbig et al., 2003; Nehrkorn et al., 2010; Pillai et al., 2011; Maier et al., 2022). The STILT model utilises the mean advection scheme used by the HYSPLIT model (Stein et al., 2015). The turbulent motions are modelled as a Markov chain process (Lin et al., 2003). The mean wind is represented by interpolating the wind fields from numerical weather prediction models or reanalysis data (from the WRF model in this study) into the sub-grid location of the particle. STILT simulates the transport by following the backwards-in-time evolution of an ensemble of particles (representing air parcels of equal mass) from receptor locations using mean winds and turbulent motions. The most critical meteorological variables required for trajectory calculations are vertical profiles of horizontal and vertical wind components (Nehrkorn et al., 2010).

In the STILT model, changes in the atmospheric $CO_2$ concentration, $\Delta C(\mathbf{x}_r, t_r)$ at the observation site at $\mathbf{x}_r$ and time $t_r$ can be derived as follows:

$$\Delta C(\mathbf{x}_r, t_r) = \int_{t_0}^{t_r} dt \int_V dx\,dy\,dz\,I\,(\mathbf{x}_r, t_r|\mathbf{x}, t)S(\mathbf{x}, t) \tag{1}$$

where $S(\mathbf{x}, t)$ is a volume source-sink in units of $ppm\ h^{-1}$ and $I\,(\mathbf{x}_r, t_r|\mathbf{x}, t)$ is the influence function for the receptor location which quantitatively links sources/sinks to concentrations and has a unit of $m^{-3}$. The quantification of the time volume integration of the influence function is achieved by counting the total length of time $\Delta t_{p,m,i,j,k}$ that each released particle $p$ spends in a volume element $(i, j, k)$ during a time step $m$ and normalising to the number of particles released $N_{tot}$ (Lin et al., 2003).

$$\int_{t_m}^{t_m+\tau} \int_{x_i}^{x_i+\Delta x} dx \int_{y_j}^{y_j+\Delta y} dy \int_{z_k}^{z_k+\Delta z} dz\,I\,(\mathbf{x}_r, t_r|\mathbf{x}, t) = \frac{1}{N_{tot}} \sum_{p=1}^{N_{tot}} \Delta t_{p,m,i,j,k} \tag{2}$$

The link between surface fluxes $F(x, y, t)$ (in units of $mol\ m^{-2}s^{-1}$) and a volume source-sink $S(\mathbf{x}, t)$ is established by diluting the surface tracer flux into an atmospheric column of height $h$, in the assumption that the turbulent mixing below this height is strong enough to thoroughly mix the surface flux from ground to $h$ within one model time step $m$. Here, $h$ is set to half of the PBL height, and the PBL height is calculated internally by STILT using meteorological inputs provided by WRF. This approach is summarised as follows:

$$S(\mathbf{x}, t) = \begin{cases} \frac{m_a ir}{h\overline{\rho}(x,y,t)}F(x, y, t) & for\ z \leq h \\ 0 & for\ z > h \end{cases} \tag{3}$$



where $m_{air}$ is the molar mass of air and $\overline{\rho}(x, y, t)$ is the average air density. From the above equations (Eqs. (1), (2) and (3)), the contribution of emission fluxes from each surface grid cell $(i, j)$ and time step $m$ to the total $CO_2$ enhancement $\Delta C(\mathbf{x}_r, t_r)$

at receptor location can be obtained as:

$$\Delta C_{m,i,j}(\mathbf{x}_r, t_r) = \frac{m_{air}}{h\overline{\rho}(x_i, y_i, t_m)} \frac{1}{N_{tot}} \sum_{p=1}^{N_{tot}} \Delta t_{p,m,i,j,k} F(x_i, y_i, t_m) = f(\mathbf{x}_r, t_r | x_i, y_i, t_m) F(x_i, y_i, t_m) \quad (4)$$

Here, $f(\mathbf{x}_r, t_r | x_i, y_i, t_m)$ is known as the "footprint" which links the $CO_2$ surface fluxes to $CO_2$ concentration changes at the observation site as mentioned before. The total $CO_2$ concentration enhancement $\Delta C(\mathbf{x}_r, t_r)$ at the observation site is obtained by summing $\Delta C_{m,i,j}(\mathbf{x}_r, t_r)$ over all the grid cells $(i, j)$ and time $(m)$.

We released 100 particles from every receptor location to calculate the back trajectories with a maximum backward time of 120 h. This period is set by estimating the approximate time required for all particles to exit the model domain. We used the time-averaged, mass-coupled velocity fields from the WRF model to avoid mass violation in STILT. The initial and boundary conditions for WRF are obtained from the ERA5 reanalysis dataset of the European Centre for Medium-Range Weather Forecasts (ECMWF). The WRF simulations over the domain are generated for 2017. The detailed description of the WRF model set

up over the Indian domain used for this study can be obtained from Thilakan et al. (2022). The performance of the WRF model simulations over India was assessed by previous studies (e.g., Hariprasad et al., 2014; Boadh et al., 2016; Sivan et al., 2021; Mathew et al., 2023), which found it promising. The footprints were calculated based on Eq. (4), which were dynamically gridded to a maximum resolution of 10 km × 10 km.

The biosphere flux distribution over the domain was generated using a biospheric model called Vegetation Photosynthesis

and Respiration (VPRM) model (Mahadevan et al., 2008). The VPRM model calculates Gross Ecosystem Exchange (GEE) and ecosystem respiration ($R_{eco}$) using WRF meteorological fields and MODIS (Moderate Resolution Imaging Spectroradiometer) satellite products. Biospheric fluxes are generated at horizontal resolution 10 km × 10 km over the domain. These fluxes were utilised to calculate the atmospheric $CO_2$ contribution by the biosphere over the receptor locations (termed as $CO_{2_{bio}}$). Anthropogenic $CO_2$ fluxes were prescribed from three different inventories to represent anthropogenic contribution and also

to examine the impact of emission differences in $CO_2$ simulations over the Indian domain. STILT derives the atmospheric $CO_2$ enhancement due to anthropogenic fluxes ($CO_{2_{ant}}$) at receptor locations using Eq. (4). Anthropogenic emission fluxes from Emissions Database for Global Atmospheric Research (EDGAR), Open-source Data Inventory for Anthropogenic $CO_2$ (ODIAC) and Integrated Carbon Observation System - Global anthropogenic $CO_2$ emissions (hereafter referred to as ICOS) were used in this study. The EDGAR inventory (v7.0; Crippa et al., 2018, 2022) provides anthropogenic fluxes at a horizontal

resolution of 0.1°×0.1°for every year. ODIAC (v2020; Oda and Maksyutov, 2020; Oda et al., 2018) has a higher spatial resolution of 1 km × 1 km but is available only at a monthly timescale. ICOS (v2019; Karstens et al., 2019; Janssens-Maenhout et al., 2019) is developed based on EDGAR v4.3 and British Petroleum statistics with a horizontal resolution of 0.5°×0.5°and a temporal resolution of hourly. All these data sets were interpolated into model resolution, conserving mass. The model included the effect of global $CO_2$ variability over the domain from boundary conditions (also known as background signal,

$CO_{2_{bck}}$) and was added to the local $CO_2$ mole fraction (resulting from local fluxes) within the model domain to compare with the observations. In this study, we have used two different global reanalysis products separately as boundary conditions to



understand the influence of boundary conditions on total $CO_2$ mole fraction. We used Jena CarboScope (version: s10c_v2020; Rödenbeck et al., 2003) and ECMWF-Copernicus Atmosphere Monitoring Service (CAMS, Version: EGG4; Agustí-Panareda et al., 2023) as boundary conditions for this study (see Sect. 3.2).

That is, the total atmospheric $CO_2$ concentration (atmospheric $CO_{2,tot}$) was calculated by adding the background ($CO_{2,bck}$), biospheric ($CO_{2,bio}$) and anthropogenic ($CO_{2,ant}$) terms together to compare with the observations. The atmospheric $CO_2$ concentration at the measurement location is given by:

$$CO_{2_{tot}} = CO_{2_{bck}} + CO_{2_{bio}} + CO_{2_{ant}} \tag{5}$$

Atmospheric $CO_2$ mixing ratios at the measurement locations were retrieved at a temporal resolution of three hours. Since

we used two different boundary conditions and three different anthropogenic fluxes for the WRF-STILT simulations, we have a set of six simulations over each observation site. WRF-STILT simulations are hereafter referred to as simply STILT simulations in this article. The simulations with CarboScope (CS) as background are represented as STILT-CS-EDG (EDGAR as anthropogenic flux), STILT-CS-ICOS (ICOS as anthropogenic flux) and STILT-CS-ODI (ODIAC as anthropogenic flux) in this manuscript. Similarly, simulations with CAMS EGG4 (EGG4) as background are represented as STILT-EGG4-EDG (EDGAR

as anthropogenic flux), STILT-EGG4-ICOS (ICOS as anthropogenic flux) and STILT-EGG4-ODI (ODIAC as anthropogenic flux).

## 3   Data

### 3.1   $CO_2$ observations over India

We used atmospheric $CO_2$ observations for 2017 from four measurement sites located at Mohali, Nainital, Shadnagar and

Nagpur (see Fig. 1) to assess their temporal variability. Also, we examine how well the STILT simulations capture these variations.

    We used continuous hourly measurements of $CO_2$ using the PICARRO CRDS (Cavity Ring-Down Spectroscopy) instrument at the Mohali (MHL) station. Atmospheric $CO_2$ mole fractions are measured at 20 m height above ground level. The measured $CO_2$ mixing ratios have an overall uncertainty calculated based on the root mean square propagation of individual uncertainties,

such as the accuracy error of gas standard (2%), $2\sigma$ instrumental precision error (0.1% for $CO_2$), and flow reproducibility (2%), resulting in a measurement uncertainty less than 4%. The limit of detection for $CO_2$ is reported to be better than 0.5 ppm (Chandra et al., 2017). MHL is situated in a suburban area (30.67° N, 76.73° E; 310 m a.s.l) in the northwestern part of the Indo-Gangetic plain (IGP), close to Chandigarh city (Sinha et al., 2014; Pawar et al., 2015). The instrument facility is housed inside the campus of the IISER Mohali. More details about the measurement techniques employed for MHL observations are

available from Chandra et al. (2017). MHL has the proximity of three cities with more than 100,000 population at a distance of a few kilometers in the northeast direction, including Chandigarh, with nearly one million population. STILT footprints show that the predominant wind direction towards the observation site is northwest, except during the monsoon season, in which the wind comes in the southeast direction (Fig. S1). The northwest region of the MHL is dominated by agricultural and other rural




land use patterns (Kumar and Sinha, 2021). Agricultural emission activities like residue burning can be expected in this region
during April-May and October-November period (Sinha et al., 2014). Local influences on the measurements from the residents
of IISER Mohali are expected to be minimal since the instrumentational facility is situated in the downwind direction.

Weekly flask measurements of atmospheric $CO_2$ mole fractions from the Nainital (NTL) observation site are used here
(Terao et al., 2022). NTL observation site is located at the Aryabhatta Research Institute of Observational Sciences (ARIES)
(29.36° N, 79.46° E; 1940 m a.s.l, Nomura et al. (2021)). Since the measurement location is near the Himalayan mountain
range, NTL is considered as a background site representing north Indian GHG distribution with some influence from anthro-
pogenic activities, including biomass burning during spring and autumn months when air mass stays for a longer duration over
northern India (Sarangi et al., 2014; Nomura et al., 2021). The inlets for the air samples are mounted at a height of 7 m above
ground level. Weekly flask samples were collected at 14:00 local time and transported to the Center for Global Environmental
Research (CGER) laboratory, National Institute for Environmental Studies (NIES), Japan, for gas analyses. $CO_2$ analyses were
done using a non-dispersive infrared analyzer (NDIR; LI-COR, LI-6252) with an analytical precision of 0.03 ppm against the
NIES09 scale and the NIES09 and NOAA scales have a difference ranging from 0.04 to 0.09. More details are available at
Nomura et al. (2021). Near-field contributions of the NTL station are mainly from the northwestern region of the station except
during the summer monsoon (JJA) period (Fig. S2). During the winter period (DJF), the influence region covers the southeast
of the site as well.

The observation site Shadnagr (SDN) is at the National Remote Sensing Sensing Center (NRSC), Shadnagar (17.09° N,
78.21° E; 648m a.s.l). Shadnagar is a suburban station situated about 65 km from Hyderabad (Mahesh et al., 2015; Sreenivas
et al., 2016). Measurements are carried out using Los Gatos Research's Greenhouse Gas Analyser (model: LGR-GGA-24EP) at
an interval of 1 s with precision and accuracy of 0.078 ppm and 0.101 ppm respectively (Mahesh et al., 2015; Sreenivas et al.,
2016). The LGR-GGA instrument uses enhanced off-axis integrated cavity output spectroscopy (OA-ICOS) technology. A
downward-facing inlet is mounted 10 m above ground level to provide ambient airflow to the instrument. Mahesh et al. (2015)
provides a detailed description of the instrument and the calibration procedure. This study uses the daily average values of
these observations available from https://bhuvan-app3.nrsc.gov.in/data/download/index.php (last access: 12 December 2022).
The near-field influence regions of SDN vary with seasons (Fig. S3). The influence region covers the northeast of the site
during post-monsoon (SON) and winter (DJF) seasons. The dominant influence on the SDN comes from the west during the
summer monsoon period (JJA) and from the southeast during the pre-monsoon season (MAM).

We have also used continuous atmospheric $CO_2$ measurements from Nagpur (NGP) installed at NRSC, Regional Centre
office (21.15° N, 79.15° E; 312 m a.s.l). NGP is located 7 km west of Nagpur city centre, one of the largest cities in central
India, with a population of around 2.5 million. The site's region (Deccan plateau of the Indian peninsula) includes large
industries and coal-powered power plants (Kompalli et al., 2014; Shaeb et al., 2020). Based on our STILT footprints, the major
influence on the $CO_2$ variability at NGP comes from the west (summer, JJA), northeast (post-monsoon, SON), and north-
west (pre-monsoon, MAM) of the observatory (Fig. S4). NGP utilises a high-precision non-dispersive infrared gas analyzer
(LICOR LI-7500) instrument mounted at 8 m height from ground level to measure the atmospheric $CO_2$ concentrations. Daily
average values of these measurements, available from https://bhuvan-app3.nrsc.gov.in/data/download/index.php, (last access:



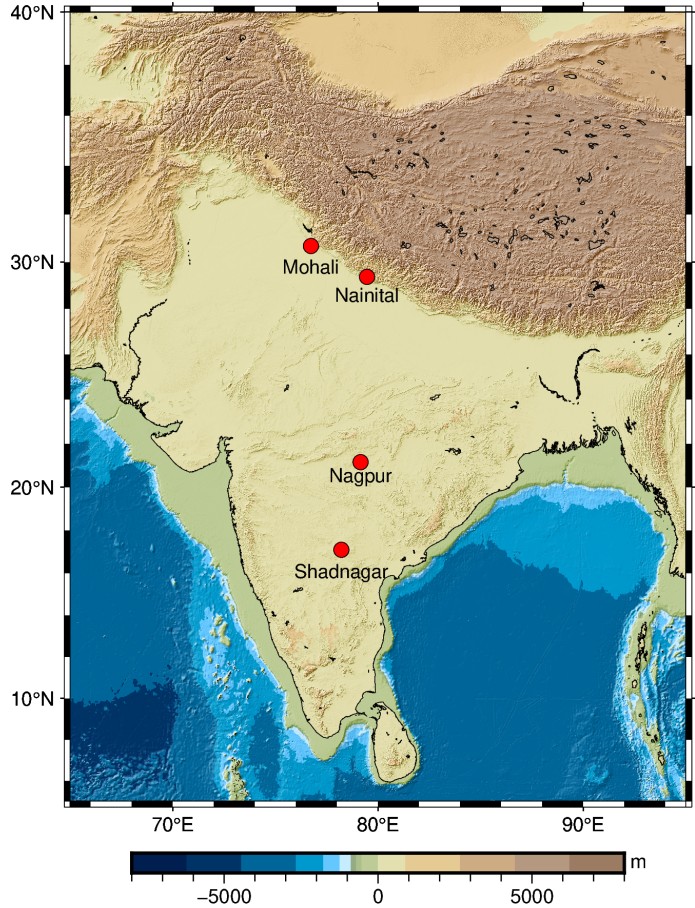

**Figure 1.** Location of $CO_2$ observation sites used in the study

12 December 2022), are used in this study. Both SDN and NGP observations are carried out as part of the Climate and
Atmospheric Processes of the ISRO-Geosphere Biosphere Programme (CAP-IGBP).

### 3.2 Global Reanalysis products

We also compared $CO_2$ observations with three global reanalysis products to examine the model-data mismatches at these
stations. These products are optimized with available observations of $CO_2$ (e.g. data from surface monitoring stations, total
column retrievals from satellites, aircraft missions, ship cruises, and AirCore balloon sounding) from different parts of the
world. None of these products utilises in-situ observations from India. We used atmospheric $CO_2$ concentration from Car-
bonTracker (CT2019B; Jacobson et al., 2020), CarboScope (s10c_v2020; Rödenbeck et al., 2003; CarboScope, 2020) and
ECMWF CAMS (EGG4; Agustí-Panareda et al., 2023; Copernicus Atmosphere Monitoring Service, 2021) to compare with
the observations. All of these reanalysis products differ in their spatial and temporal resolutions. CarbonTracker has a horizon-





tal resolution of $3° \times 2°$ and a temporal resolution of 3 hours with 25 vertical levels. CarboScope is a comparatively coarser
model with a horizontal resolution of $5° \times 3.8°$ and a temporal resolution of 6 hours with 19 vertical levels. Among these prod-
ucts, CAMS EGG4 has the finest spatial resolution with $0.75° \times 0.75°$ in the horizontal direction with 25 vertical levels and a
temporal resolution of 3 hours. To compare with the MHL, SDN and NGP observations, we used global model simulations
from the model's first vertical level for CarbonTracker and the 1000 mb pressure level for both CarboScope and EGG4 prod-
ucts. Since the NTL site is situated at $\sim$ 800 mb height, we compared the observations with CarboScope and EGG4 products
at 800 mb vertical level and with the first model level for CarbonTracker.

## 4  Assessment of modelling skill

We have derived different statistical indices to examine the performance of the model simulations to predict the $CO_2$ variability.
To quantify the error distribution between model ($P$) and the observation ($O$), we have calculated the Root Mean Square Error
(RMSE) and the Mean Absolute Error (MAE) between the model simulations and observations.

To separate the systematic and unsystematic components from the RMSE, we have used the following method proposed by
Willmott (1981). Systematic RMSE is obtained as:

$$RMSE_s = \sqrt{\frac{1}{n}\sum_{i=1}^{n}(\hat{P}_i - O_i)^2} \tag{6}$$

and the unsystematic RMSE as:

$$RMSE_u = \sqrt{\frac{1}{n}\sum_{i=1}^{n}(P_i - \hat{P}_i)^2} \tag{7}$$

where $\hat{P}_i = a + bO_i$

Here $a$ and $b$ respectively are the intercept and slope of the least squares regression. The systematic difference for a 'perfect'
model is expected to be very close to zero, while the unsystematic difference remains close to the value of RMSE. Based on
Willmott et al. (2012), we have also computed the refined index of agreement ($d_r$) as follows:

$$d_r = \begin{cases} 1 - \dfrac{\sum_{i=1}^{n}|P_i - O_i|}{c\sum_{i=1}^{n}|O_i - \overline{O}|}, & \text{when } \sum_{i=1}^{n}|P_i - O_i| \leq c\sum_{i=1}^{n}|O_i - \overline{O}| \\[4mm] \dfrac{c\sum_{i=1}^{n}|O_i - \overline{O}|}{\sum_{i=1}^{n}|P_i - O_i|} - 1, & \text{when } \sum_{i=1}^{n}|P_i - O_i| > c\sum_{i=1}^{n}|O_i - \overline{O}| \end{cases} \tag{8}$$

Here the constant c is set to 2 (Willmott et al., 2012). The $d_r$ values can range from -1 to 1. It indicates the sum of error
magnitudes between predicted and observed values relative to the sum of observed deviations around the observed mean. For





example, $d_r$ = 0.5 indicates that the sum of the model-observation mismatch is half the sum of the observed variability around the mean. i.e. $d_r$ gives a relative magnitude of the model error compared to the variance of the observations.

## 5 Results

### 5.1 Observed CO$_2$ variability over India

To assess the CO$_2$ variability over India during 2017, we analysed in-situ observations of atmospheric CO$_2$ from four different sites (see Sec. 3.1).

MHL observations show strong variability due to its proximity to urban areas (see Fig. 2). MHL has hourly observations, and we have separated the daytime values (11:00-16:00 Local time) to distinguish the influence of the nocturnal boundary layer on observations (see Fig. 2). The annual mean of hourly atmospheric CO$_2$ concentration at MHL (whole day) during 2017 is 428.8 ppm with a standard deviation ($\sigma$) of 26.6 ppm. For the daytime, the annual mean CO$_2$ is approximately 20 ppm less (408.3 ppm) than all-time, with a variability ($\sigma$) of 11.6 ppm. Observations show strong diurnal variability ($\sigma$ = 14.7 ppm) with up to 40 ppm difference between the maximum and minimum concentrations during the early morning (06:00 LT) and the afternoon (15:00 LT), respectively (Fig. S5). Due to strong mixing, variability in CO$_2$ concentration is less ($\sigma \approx 12$ ppm) during daytime (12:00-15:00 LT) compared to the nocturnal variability of 19-32 ppm (see Fig. 3). A similar reduction in CO$_2$ variability can be seen at 09:00 LT during May-August (Fig. S6) due to a well-established convective boundary layer with strong mixing. Nocturnal CO$_2$ variability during March-May is less compared to other seasons. Other than this, MHL observations do not show considerable differences in their diurnal cycle among seasons (Fig. S6). Since most of the inverse models, which target to retrieve surface-atmosphere exchange fluxes from in-situ observations, use daytime measurements, we carry out the rest of our analysis for MHL based on daytime values. Monthly mean values of daytime observations show that MHL exhibits strong seasonal variability ($\sigma$ = 9.2 ppm) with approximately a 32.9 ppm difference between maximum and minimum values (see Fig. S7). Maximum intra-month variability is found during January, September, November and December, with a standard variability of 8-12 ppm (Fig. 2). On a monthly scale, lower values are seen during February (397.9 ppm) and August (391.2 ppm) and higher values during May (413.5 ppm) and November (424 ppm) (Fig. S7). As expected, the atmospheric CO$_2$ concentration decreases from June onwards due to the enhanced biospheric activity associated with summer monsoon rainfall (Fig. S7). However, we find high CO$_2$ concentrations at MHL during November, which can be attributed to the agricultural waste-burning activities prominent around this region at this time of the year. A detailed discussion on the influence of biomass burning on CO$_2$ concentration over MHL is provided in Sec 6.1. In general, measured CO$_2$ concentration over MHL shows considerable influences from local fluxes (see Fig. S8).

Weekly observations from NTL also show strong seasonal variations ($\sigma$ = 7.5 ppm) in CO$_2$ concentrations (Fig. 4) with a difference of up to 25 ppm, between the maximum (412.8 ppm) and minimum (387.9 ppm) concentrations during April and September respectively. The observations show an annual mean of 401.6 ppm with a variability reaching 8.4 ppm during 2017. Considerable intra-month variations at NTL are observed during August, October and December with the variability of $\sim$ 6



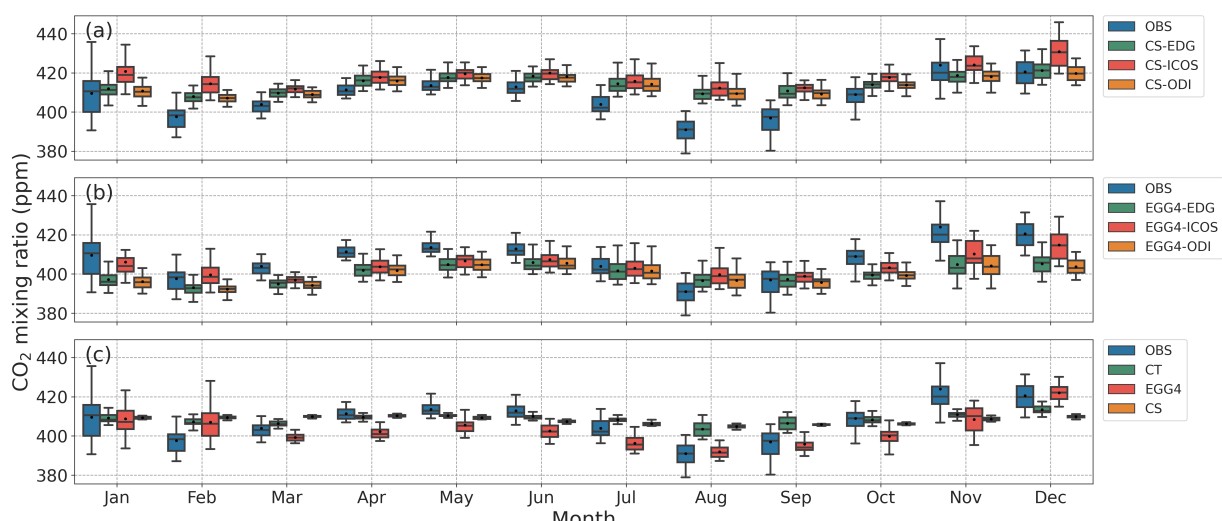

**Figure 2.** $CO_2$ monthly variations over MHL during 2017. Observed $CO_2$ variability during daytime (11:00 - 16:00 local time) is shown in comparison with (a) STILT-CS simulations, (b) STILT-EGG4 simulations and (c) global reanalysis products.

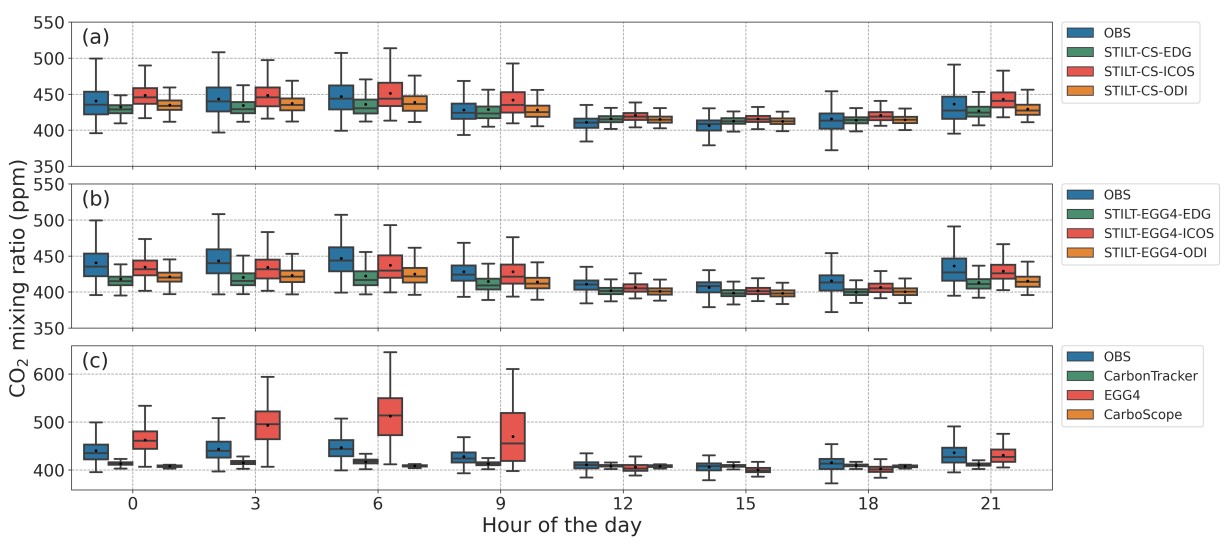

**Figure 3.** $CO_2$ diurnal cycle over MHL during 2017 is shown in comparison with (a) STILT-CS simulations, (b) STILT-EGG4 simulations, and (c) global reanalysis products. Note that STILT provides output only every three hours. Similarly, EGG4 and CarbonTracker provide outputs at a three-hour resolution and CarboScope at a six-hour resolution.



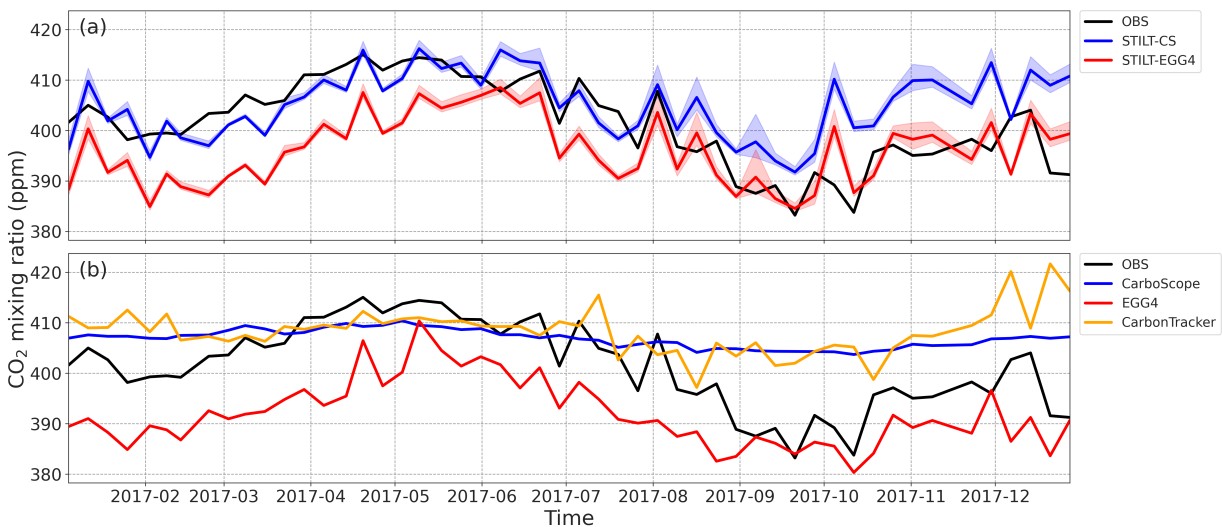

**Figure 4.** $CO_2$ time series of weekly observations (14:00 local time) at NTL with (a) STILT simulations. Blue (STILT-CS) and red (STILT-EGG4) curves represent the ensemble average of the STILT simulations using different anthropogenic fluxes. Shaded regions represent the range of the model simulations. (b) Global reanalysis products.

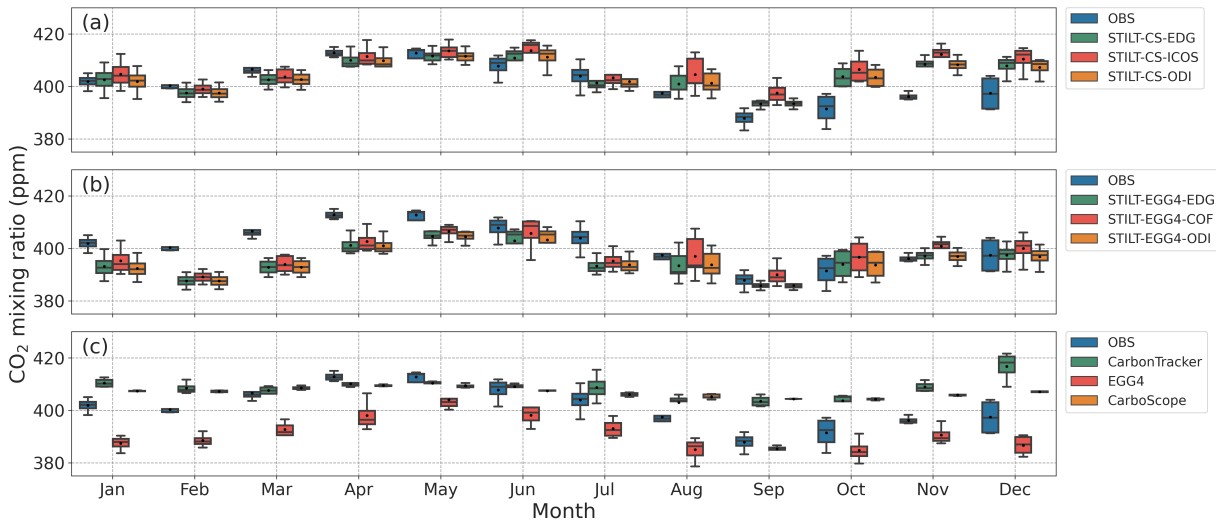

**Figure 5.** $CO_2$ monthly variations over NTL during 2017. Observed $CO_2$ variability is shown in comparison with (a) STILT-CS simulations, (b) STILT-EGG4 simulations and (c) global reanalysis products.

ppm (Fig. 5). In August, $CO_2$ concentrations show a sharp decrease in the concentration of $\sim 18$ ppm from previous values at
the beginning of the month ($\sim 408$ ppm, see Fig. 4).



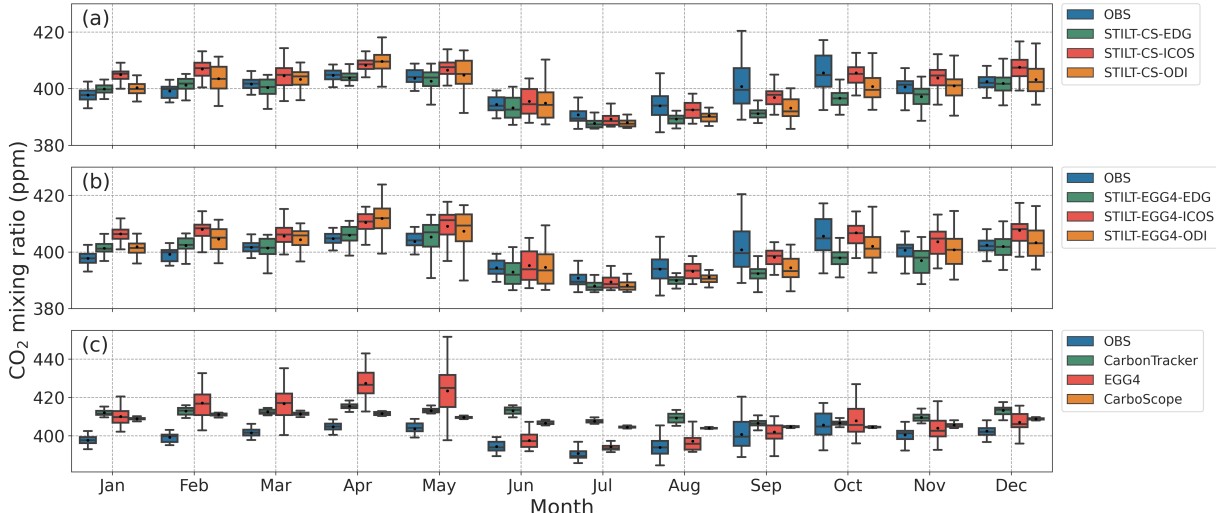

**Figure 6.** $CO_2$ monthly variations over SDN during 2017. Observed $CO_2$ variability is shown in comparison with (a) STILT-CS simulations, (b) STILT-EGG4 simulations and (c) global reanalysis products.

The annual mean of daily SDN observations is 399.6 ppm during 2017 with a standard deviation of 6.2 ppm (Fig. 6). At SDN, measurements show a seasonal $CO_2$ variability of 4.4 ppm, with two peaks during April (404.7 ppm) and October (405.6 ppm). The lowest concentration is observed during July (390.8 ppm), with a difference of 14.8 ppm from the highest monthly concentration (Fig. S9). In general, SDN observations have not shown much intra-month variability ($\sigma \approx 2$ ppm, see Figs. 6 & S10) except during the period from August to October ($\sigma$ ranges from 5.2 to 8.3 ppm). Only daily mean observations from SDN are available for analysis, not hourly data as desired.

The $CO_2$ measurements at NGP during 2017 show an annual mean of 415.2 ppm, with a variability of 9.5 ppm (Fig. 7). NGP observations show seasonal variability ($\sigma = 7.7$ ppm) with two maxima and one minimum value with $\sim 22$ ppm difference between these peaks (Fig. S11). Enhanced $CO_2$ concentrations are observed during May (426.0 ppm) and October (425.6 ppm). In July, the NGP observations show lower concentrations (404.14 ppm) than the rest of the period. A sharp reduction of $CO_2$ concentration ($\sim 13$ ppm) is found from October to December (see Figs. S11-S12). Mostly NGP $CO_2$ observations indicated $\sim$ 4 ppm variability within a month (see Figs. 7 & S14), except in June (6.4 ppm), September (9.8 ppm) and October (7.6 ppm). Here also, we only have access to daily mean observations from NGP, not to hourly data.

## 5.2 Comparison between Observations and WRF-STILT model Simulations

We assessed how well the STILT model simulations agree with observed $CO_2$ variability. For the comparison, we used observations from all four stations described in Sect. 3.1 and a set of six STILT $CO_2$ simulations (see Eq. 5) as described in Sect. 2.





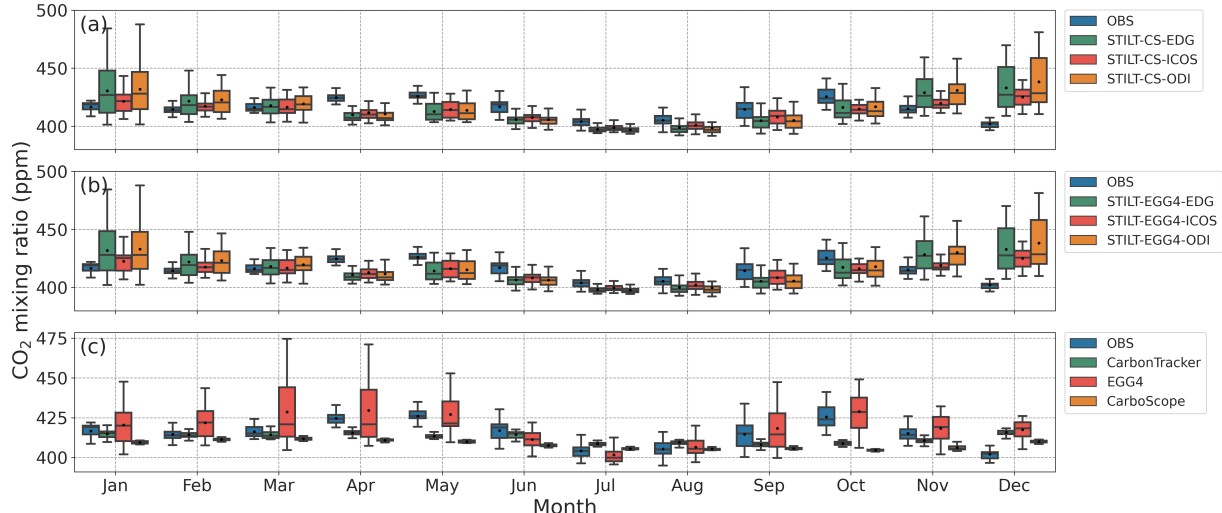

**Figure 7.** $CO_2$ monthly variations over NGP during 2017. Observed $CO_2$ variability is shown in comparison with (a) STILT-CS simulations, (b) STILT-EGG4 simulations and (c) global reanalysis products.

Figure 2 (see a-b) shows the comparison of MHL daytime observations during 2017 with the STILT simulations. Overall, STILT simulations capture the observed daytime variations reasonably well, with slight overestimation for STILT-CS simula-

tions and slight underestimation for STILT-EGG4 simulations. Similar to observations, STILT simulations during March-July show less intra-month variability. The maximum variability is found during the winter months. A detailed discussion on the differences in $CO_2$ simulations while using EGG4 and CarboScope as the initial and boundary conditions is provided in Sect. 6.3. STILT simulations with ICOS anthropogenic fluxes showed higher variability ($\sigma \approx 7.3$ ppm) than the other simulations.

Though STILT simulations capture the seasonal $CO_2$ variability in monthly averaged daytime values over MHL (see Fig.

S7), the models failed to represent a sharp decline in $CO_2$ concentration during December. This may be due to the increased biospheric uptake by Rabi crops during this period (see Sect. 6.2). At the same time, monthly averaged values of daytime observations show a second dip in February, which is captured reasonably well by the STILT simulations (Fig. S7). The simulations could reasonably reproduce the biospheric uptake in August by showing the lowest $CO_2$ concentration in August, similar to observations. The correlation coefficient between monthly averaged observations and STILT simulations varies

between 0.86 to 0.89 (STILT-CS) and 0.76 to 0.87 (STILT-EGG4). At a monthly scale, STILT-EGG4 simulations underestimate the seasonal cycle over the MHL (RMSE: 6.7 - 10.0 ppm), while STILT-CS simulations show an overestimation (RMSE: 8.2 - 11.5 ppm). The annual averaged diurnal $CO_2$ concentration shows a good correlation (see Fig. S5) between observation and STILT simulations (0.97-0.99). But there is a significant bias in the STILT-simulated diurnal cycle (see Fig. S5), which is higher for STILT-EGG4 simulations (7.0-18.5 ppm) compared to STILT-CS simulations (5.4-8.2 ppm). The estimated bias is

small during summer (MAM) compared to other seasons (Fig. S6). Observations and STILT simulations show less variability



**Figure 8.** An overview of the performances of different models (see Sect. 4). Bar plots represent the different RMSE (in teal), systematic RMSE (RMSE$_s$, in lime green) and unsystematic RMSE (RMSE$_u$, in orchid) values estimated for each station. MAE (●), observed standard deviation (✖) and model standard deviation (✚) are overlied on barplots. The black and blue lines represent the correlation coefficient and index of agreement values, respectively. (a) Mohali (b) Nainital (c) Shadnagar (d) Nagpur





during daytime (12:00-18:00) compared to other periods (Fig. 3), also showing a good model-data agreement during daytime (Figs. 3 & S5).

Figure 8 summarises the statistical indices (see Sect. 4) estimated for assessing the model skills. At the MHL station, STILT $CO_2$ daytime simulations show a standard variability (Fig. 8a) ranging from 5.3 - 7.3 ppm during 2017, lower than the observed standard variability (11.6 ppm). RMSE for STILT simulations shows a maximum of 13.6 ppm (STILT-CS-ICOS) and a minimum of 10.4 ppm (STILT-EGG4-ICOS). MAE values follow the same pattern as RMSE with reduced magnitude. STILT simulations show a reasonable correlation with observations with coefficient values ranging from 0.53 (STILT-EGG4-EDG) to 0.61 (STILT-CS-ODI). The index of agreement estimated for MHL varies from 0.44 (STILT-CS-ICOS) to 0.66 (STILT-CS-ODI), indicating that the error values have a magnitude less than or equal to the variability of observations. Analysis of these indices for different months indicates that STILT has comparatively better prediction capability in summer (March-June) than the rest of the period (see Additional Material, Fig. AM5). The above results show the models' difficulty reproducing mixing during monsoon and winter. An inadequate representation of biospheric flux activities in the model can also result in observation-model mismatches. The model skill indices estimated for November are poor owing to the likely misrepresentation of variability associated with biomass burning during these months (Sinha et al., 2014; Pawar et al., 2015).

At NTL, STILT simulations captured the $CO_2$ variability reasonably well, except in the winter period (Figs. 4-5). An offset of 5 ppm is used in STILT-CS simulations at NTL to minimize the consistent overestimation by the model (i.e. 5 ppm is subtracted from the initial $CO_{2bck}$ component). A sharp reduction in observed $CO_2$ concentration from August (Fig. 4) was not captured by the models (see Sect. 6.2 for a detailed discussion). Noticeably STILT-EGG4 simulations showed an underestimation of $CO_2$ values from January to May. The simulations have a standard deviation of $\sim 6$ ppm (6.2 - 7.1 ppm) in $CO_2$ concentration during 2017 which is lower than the observed standard deviation (Fig. 8b). The RMSE estimated for STILT simulations over NTL varies between 7.3 to 9.0 ppm. STILT simulations show a reasonable correlation with the observations with a correlation coefficient of $\sim 0.6$, except for simulations using ICOS anthropogenic emission fluxes. We get an index of agreement of $\sim 0.5$, indicating that the magnitude of STILT model error is half that of the observed variations about the observed mean at NTL.

Comparison of $CO_2$ observations with the model simulations at SDN shows that the STILT models can predict the seasonal cycle very well (see Figs. 6 & S9) with an RMSE of $\sim 4$ ppm and correlation ranging from 0.75 to 0.87. Like NTL, we reduced an offset of 20 ppm in STILT-CS simulations and 5 ppm in STILT-EGG4 simulations to correct the initial $CO_{2bck}$ component. STILT reasonably reproduces the observed intra-month variability except from August to October (Figs. 6 & S10). For SDN, the standard deviation of STILT simulations is higher than the observations (6.2 ppm) and ranges from 6.2 to 8.5 ppm. Estimated RMSE values for STILT simulations are comparatively low at SDN and range from 6.2 to 7.2 ppm (Fig. 8c). MAE values vary from 4.3 to 5.5 ppm and follow a similar pattern as RMSE. STILT simulations show a reasonable correlation (0.55-0.67) with the observations at SDN. But the index of agreement is close to zero for two simulations (STILT-EGG4-ODI and STILT-EGG4-ICOS), indicating a model error in the simulations as high as observational variability. All STILT simulations show less model skill from August to November. STILT-EGG4 simulations show comparatively less model skill during January-May (see Additional material Fig. AM6).



STILT simulations at NGP capture the observed seasonal variability except for the winter season (Figs. 7 & S11). The models represent the seasonal cycle from March to October over NGP with a correlation coefficient of $\sim 0.97$ and RMSE of $\sim$ 9 ppm. We reduced an offset of 15 ppm in STILT-CS simulations. The STILT simulations overestimate the winter variability. Also, the intra-month variability at NGP is overestimated during winter (Figs. 7 & S12). Notably, the observed decrease in $CO_2$ concentration during the summer monsoon season is well-captured by STILT (Figs. 7 & S11). But the increase in the $CO_2$ concentration in STILT simulations during winter months (November-February) is absent in $CO_2$ observations over NGP (Fig. S11). The skill indices for the NGP station show that the standard deviation of STILT simulations (10.1-17.9 ppm) is higher than observed standard deviations (Fig. 8d). Also, higher RMSE (10.5-17.5 ppm) are estimated for STILT simulations at NGP. Model simulations show very poor correlation coefficient values and index of agreement values at the NGP region for 2017. We obtained a better observation-model agreement when excluding winter months (November-February) (see Additional material Fig. AM7). Analysis of model skill indicates that the June-August period has low RMSE, which can be associated with strong mixing by monsoon winds (see Additional material Fig. AM8).

## 5.3 Comparison between Observations and Global Reanalysis products

We compared observations with three global reanalysis products (CarbonTracker, CarboScope and EGG4) described in Sect. 3.2. The global reanalysis products (except EGG4) could not capture the seasonal variability in $CO_2$ over MHL (Fig. S7). The intra-month variability is less in daytime simulations of global models (Fig. 2) except for EGG4 simulations during winter months (November-February). CarbonTracker and CarboScope show much lower seasonal and intra-seasonal variability over MHL (Fig. S7). CarbonTracker exhibit diurnal $CO_2$ variability with a significant underestimation (Fig. S5). EGG4 captured the diurnal variability reasonably well, with a considerable nocturnal bias (Fig. S5). Note that the EGG4 has the highest spatial and temporal resolution among the global reanalysis products used in this study. Also, long-range transport has a strong influence on the MHL site (Pawar et al., 2015), which might be contributed to EGG4's better performance. The inter-model differences in intraseasonal variability are large over MHL (Fig. 2), with the standard deviation ranging from 1.9 (CarboScope) to 9.8 ppm (EGG4). The RMSE for global model simulations varies from 10.2-12.0 ppm with correlation coefficients ranging from 0.36-0.52 (Fig. 8a). While we consider the index of the agreement, EGG4 show lower values (0.47) compared to other products, indicating that the magnitude of the error is approximately half of the observed variations.

CarboScope and CarbonTracker also did not capture the seasonal variability in $CO_2$ concentration at NTL. Though it underestimated the variability, EGG4 showed good agreement with the seasonal variations in observations (see Fig. 4). These reanalysis products show significant differences in $CO_2$ variability with standard deviations varying as 1.8 ppm (CarboScope), 4.4 ppm (CarbonTracker) and 6.9 ppm (EGG4). The observed standard deviation was higher than the standard deviation in these products (Fig. 8b) except for EGG4. EGG4 has the maximum RMSE (11.4 ppm) among all products. CarbonTracker also shows a higher RMSE with 10.6 ppm than CarboScope (8.8 ppm). CarboScope and EGG4 models show high correlation values ($\sim 0.8$) compared to CarbonTracker simulations. EGG4 has a very low index of agreement compared to other model simulations, which indicates that the error in the model simulations is very high compared to the observed variability.



The standard deviation of EGG4 (12.4 ppm) simulations is higher than the observed standard deviation at SDN. But Carbon-Tracker and CarboScope predicted variability lower than that of the observation (Fig. 8c). EGG4 captured the seasonal cycle
over SDN reasonably well but shows high positive bias reaching up to ∼ 20 ppm during January-May (Fig. S9). CarbonTracker and CarboScope could not capture the seasonal variability over SDN. The intra-month variability was also poorly represented (Fig. S10) by these products. EGG4 shows the highest RMSE (13.7 ppm) among the global models. The correlation of reanalysis products with SDN observations is low (0.24-0.32) except for EGG4 (0.56) simulations. But the index of agreement is less than zero for the EGG4 product, indicating the presence of noise in the simulations.

Among global models, EGG4 shows better agreement with the observations at NGP though the variability compared to observations is very high (Fig. 7). The decline in $CO_2$ concentration during the summer monsoon season is captured by the EGG4 model (Figs. S11-12). EGG4 shows good agreement with the monthly averaged observations at NGP. But CarbonTracker and CarboScope do not capture the variability in the seasonal cycle well. The standard deviation of EGG4 (14.7 ppm) at NGP is higher than the observed standard deviation (Fig. 8d). But the standard deviations of CarboScope (2.7 ppm) and
CarbonTracker(3.4 ppm) are lower than the observed standard deviation. Estimated RMSE at NGP varied from 9.3 to 11.2 ppm. Similar to STILT simulations, global model simulations also show very poor correlation coefficient values at NGP for 2017.

## 6 Discussions

Here we further explore the shortcomings that need to be addressed to use potential $CO_2$ observations from India for inverse
optimization. Three of the four observation sites used in this study (MHL, SDN, NGP) are situated near cities and are characterised by large intra-seasonal variability. Observations from all these four sites show strong seasonal variations in $CO_2$ concentrations (see Sect. 5.1). Typically, the highest $CO_2$ concentrations are observed during the April-May period and the lowest values are observed during July-September. The seasonal decrease in $CO_2$ concentration is associated with the increase in biospheric uptake owing to monsoon rainfall (see Sect. 6.3). The seasonal progression of the biospheric uptake across India
can also be seen in the observations. That is, observations from the northern part of India (MHL, NTL) show the seasonal troughs in $CO_2$ concentrations approximately one month after the seasonal troughs in southern Indian stations (SDN, NGP). This time lag in ecosystem uptake for northern Indian sites is caused by the monsoon trajectories that result in different arrival times for precipitation across India. Along with the seasonal variations, these observations (except NTL) are also characterised by strong small-scale variability associated with local flux variations. It is thus challenging for current models (see Sect. 5.3)
to utilise them for inverse optimization. Using the STILT model has improved the capabilities in simulating these fine-scale variabilities. However, we must critically examine how well our modelling system can utilise these observations to deduce optimal information on underlying fluxes at different spatial and temporal scales. The major implications of our results are discussed here, with the interest of further improving the carbon data assimilation approaches.



## 6.1 Influence of biomass burning on CO$_2$ variability

Agricultural residue burning makes up the major share of biomass burning across India (e.g., Kumar et al., 2011). So the spatiotemporal extent of biomass fires over India closely follows the area and period of crop harvest. Thus, a greater extent of biomass burning is expected for pre-monsoon and post-monsoon seasons than for the monsoon season. Considerable aerosols and trace gas emissions are associated with these open agricultural residue burning in Indo Gangetic Plains (IGP) and Central India (Bhardwaj et al., 2016; Ravindra et al., 2022; Deshpande et al., 2023). For example, the CO$_2$ emission estimated from

biomass burning over Punjab (a northern Indian State in which MHL is located) is 15.62 MtCO$_2$yr$^{-1}$ for the year 2017 (Deshpande et al., 2023). Among the Indian states, Punjab has one of the highest rates of agricultural burning (Sahu et al., 2021; Ravindra et al., 2022; Vellalassery et al., 2021; Deshpande et al., 2023). Consequently, we found a considerable influence of agricultural biomass burning on observations at MHL during November 2017. Many biomass-burning activities were reported in late October and early November 2017 (https://firms.modaps.eosdis.nasa.gov, last access: 14 April 2023). Atmospheric CO$_2$

concentration increased up to 50 ppm, likely in response to the residue burning, with maximum concentration observed during 5-13 November 2017 (see Fig. S8). STILT-derived footprints (Fig. S1) during November cover the northwest region of MHL, indicating the possible influence of biomass burning on the observed variability at MHL. We find a considerable increase in the MODIS-derived fire counts (MODIS-FIRMS, 2021) for October and November 2017 over the MHL footprint region (see Fig. S13a). A sharp increase in the number of fire occurrences during late October and early November is very likely due to

agricultural waste burning after the harvest. We have conducted STILT simulations using biomass burning fluxes from Global Fire Assimilation System (GFAS) fluxes (Kaiser et al., 2012), and Fire INventory from NCAR version 2.5 (FINNv2.5) fluxes (Wiedinmyer and Emmons, 2022; Wiedinmyer et al., 2023). STILT simulations using FINN (STILT-FINN) indicate some influence from biomass burning with a time-lead with the CO$_2$ observations (see Fig. S13b). However, STILT simulations using GFAS (STILT-GFAS) could not represent the CO$_2$ contributions from biomass burning (Fig. S13b). The reanalysis products

also failed to capture these variability associated with biomass burning (see Fig. S8). Along with mixing height issues, misrepresentation of emission fluxes, as seen here, can lead to significant errors in the simulated distribution of CO$_2$. This result shows the role of high-resolution biomass burning fluxes in representing CO$_2$ variability at MHL.

## 6.2 Impact of biospheric uptake by crops on CO$_2$ variability

Biospheric fluxes determine CO$_2$ variability at NTL as indicated by the STILT simulated CO$_{2_{bio}}$ component, which shows a

similar seasonal cycle as that of the CO$_2$ observations (Fig. S14a). However, the simulated biospheric contribution changes from a negative (sink) to a positive (source) sign during October, while the model significantly overestimates the CO$_2$ concentration (Fig. 4, Fig S14a). This overestimation corresponds to the misrepresentation of biospheric uptake of about 127 % in the influence region during October-December by the VPRM model. The improved model's correlation with observations (correlation coefficient: 0.80) and RMSE values (reduced to ∼ 5 ppm) after adjusting the biospheric component (increasing

127 % of biospheric uptake during October-December in the total influence region) increases confidence in simulated transport




at NTL (see Additional material Fig. AM9-10). Thus, the above results demonstrate the potential of using NTL observations via high-resolution inverse modelling to inform about the biospheric fluxes.

The variations in $CO_{2_{bio}}$ over NTL are dominated by crop production (see $CO_2$-NEE_CROP in Fig S14a). The variability in the Normalized Difference Vegetation Index (NDVI, https://developers.google.com/earth-engine/datasets/catalog/COPERNICUS_
S2_SR_HARMONIZED, last access: 25 June 2023) in the footprint region of NTL, retrieved by Sentinel-2 Multispectral instrument (MSI) shows a similar pattern as of the NTL $CO_2$ observations (see Figs S2 and S14). The influence region of the NTL site covers the Indian states Uttarakhand, Himachal Pradesh and parts of Punjab, Haryana and Uttar Pradesh. The close association of NDVI patterns with the Kharif and Rabi cropping seasons over the region confirms the enhanced ecosystem uptake due to agricultural activities (Fig. S14). For example, NDVI increased from July, peaking at the end of August and remained high until October. The Kharif crop cultivation season usually starts in June-July, with a harvesting period from October to November. Further, NDVI increased in December, with another peak in February. Rabi cultivation typically happens around November; these crops are harvested in March-April. The lowest NDVI values for this region are associated with the harvesting period in April-May. Hence, the decrease in $CO_2$ concentration at NTL during August can be very likely due to the strong uptake of Kharif crops from the upwind locations of the NTL station. A slight decrease of NTL $CO_2$ concentration at the end of December is in response to the biospheric uptake by Rabi crops (Umezawa et al., 2016). Nomura et al. (2021) also suggests the influence of cropping patterns over IGP on NTL observations. Noticeably, the simulated $CO_2$ uptake component (contribution from Gross Primary Production (GPP)) from crops shows a similar pattern as that of the Sentinel-2 derived NDVI in the influence region of NTL (Fig. S14). At the same time, the simulated $CO_2$ component due to the crop's respiration also shows a similar magnitude of contribution as of uptake, nearly neutralising the net biospheric $CO_2$ contribution (Fig. S14a). The overestimation of STILT simulations during October-November over NTL can thus be due to the considerable overestimation of respiration fluxes or slight underestimation of carbon uptake from crops. Similarly, MHL observations are influenced by the Rabi cultivation in Punjab and IGP, showing a decrease in $CO_2$ concentration during the December-January period, which the models do not represent well. Note that the VPRM fluxes used in the current STILT simulations are not calibrated with flux observations across India. An improved prediction of biospheric $CO_2$ uptake and release can be achieved by utilising flux observations from different ecosystems to calibrate the model parameters over India (e.g., Ravi et al., 2023) and by using $CO_2$ observations in the carbon data assimilation.

### 6.3 Relative contribution of $CO_2$ components to variability

Here we discuss the contribution from different components, viz. background ($CO_{2_{bck}}$), biospheric ($CO_{2_{bio}}$) and anthropogenic ($CO_{2_{ant}}$) to the total $CO_2$ concentration.

On an annual scale, observations from MHL, SDN, and NGP sites contain contributions from local fluxes (anthropogenic and biospheric components) by approximately 6 % of the total concentration (see Fig. 9). Regionally advected signals (background component) mostly contribute (99 % of the total) to the NTL site. $CO_{2_{ant}}$ and $CO_{2_{bio}}$ show almost equal annual contributions in magnitude to the total $CO_2$ concentration in these sites. At the same time, the proportions of contributions to the total $CO_2$ can vary with seasons, such as winter (DJF), pre-monsoon (MAM), monsoon (JJA) and post-monsoon (SON) (see Additional ma-





**Figure 9.** Variability in STILT $CO_2$ emission components at different stations. For better comparison with other emission components, 380 ppm is reduced from background components. (a) Mohali daytime (11:00-16:00 local time) simulations (b) Nainital (c) Shadnagar (d) Nagpur

terial Fig. AM11-14) due to variations in atmospheric mixing and local fluxes. For instance, the reduction in $CO_{2bio}$ component over SDN and NGP during JJA can be very likely due to increased uptake and mixing during the monsoon period.

There are large differences in local fluxes that drive observed $CO_2$ variability over different sites. MHL and NGP have $CO_2$ variability dominated by anthropogenic activities ($\sigma \approx 5.4$ to 13 ppm), while most of the $CO_2$ variability at the NTL station, situated in the foothills of the Himalayas, is caused by biospheric activities ($\sigma = 5.5$ ppm)(see Fig. 9). Annually, anthropogenic
and biospheric components almost equally contribute to SDN variations ($\sigma \approx 2.9$ to 4.2 ppm).

## 6.4   Influence of emission uncertainties on $CO_2$ simulations

On estimating terrestrial carbon fluxes, inverse modelling systems usually assumes a known contribution from anthropogenic emissions. However, this assumption would be problematic when we utilise observations near urban locations which are strongly influenced by anthropogenic emissions. For instance, the mean $CO_{2ant}$ component at MHL varies as much as 4 ppm





between different emission inventories (EDGAR:3.1 ppm, ODIAC:2.5 ppm and ICOS:7.1 ppm, see Figure 9). Similarly, an emission contribution difference of up to 5 ppm, as shown by SDN and NGP simulations, also has the potential to bias the inverse flux estimations (Houweling et al., 2010; Schuh et al., 2019). At the same time, NTL shows the least differences among emission contributions (EDGAR:1.3 ppm, ODIAC: 1.3 ppm and ICOS:3.8 ppm), where the above assumption is unlikely to propagate large errors in terrestrial carbon estimations. The choice of emission inventory matters in the regional inverse systems

since they may control the majority of $CO_2$ variability when urban sites are utilized. Our results demonstrate large differences among the $CO_{2ant}$ simulations utilising different inventories, indicating the knowledge gap in the emission estimations.

## 6.5   Sensitivity of simulations to initial $CO_2$ distribution

STILT prescribes the initial concentration from global models to add the influences from the far-field fluxes to the site simulations. The spatiotemporal details in the prescribed global model can thus influence the STILT $CO_2$ simulations. The differences

between the two global reanalysis products used in our STILT simulations caused considerable inter-model mismatches in MHL (14 ppm) and NTL (9 ppm) (see Fig. 9, Background) while resulting in a negligible bias in SDN and NGP. Hence the uncertainty in representing far-field influences may cause systematic bias in simulated $CO_2$ concentrations, depending on the sites.

## 6.6   An assessment of usability of $CO_2$ in STILT-based inverse modelling

The disagreements between the observations and simulations largely arise from issues in representing mesoscale transport and local flux influences. Note that the STILT model can represent the seasonal variability at the observational sites (see Sect. 5.2) across India. Since improper accounting of $CO_2$ variability biases the inverse estimations, examining the systematic and unsystematic error terms (decoupled using Eq. (6) & (7), see Fig. 8) in STILT simulations is particularly relevant to assess the readiness of our models to utilise these measurements in the carbon assimilation system.

The $RMSE_s$ for STILT simulations over NTL varies from 5.4 to 7.6 ppm, constituting about 52-60 % of the total RMSE. However, difficulty in capturing decreased $CO_2$ associated with the enhanced biospheric contribution in NTL from August to December (more details in Sect. 6.2) indicates the inadequate representation of ecosystem uptake in the model. For instance, STILT-CS simulations could reproduce the observational variability from January to July with an $RMSE_s$ of ∼1.5 ppm. Besides the growing period (August to December), the NTL model-observation mismatch reports only 14 % of the systematic

component in the total uncertainty (see Additional material Fig. AM15). Similarly, SDN and NGP simulations resulted in $RMSE_s$ of 1.6-3.6 ppm (4-34 % of total uncertainty) and 5.1-6.5 ppm (8-34 % of total uncertainty), respectively. The above results indicate the high-resolution models' ability to utilise the observations from NTL, SDN, and NGP in inverse modelling.

However, the MHL model-observation mismatch is more systematic (66-86 %). High $RMSE_s$ values are found over MHL for most of the cases except for STILT-EGG4-ICOS simulations that utilised EGG4 products as initial & background condition

and ICOS anthropogenic emission fluxes (Fig. 8 a). These derived RMSE components and the higher percentage of systematic error contribution suggest further improvements in the models for potentially using MHL data in inversion. The EGG4 product has higher spatiotemporal resolution compared to CarboScope, which may contribute to more realistic boundary conditions





for the STILT simulation over MHL. Similarly, the ICOS inventory is the only emission flux used in this study incorporating diurnal, weekly and monthly temporal variations. A reduced $RMSE_s$ (66 %) for models using the ICOS inventory suggests a
need for representing temporal variations in emission fluxes for improved model performance at MHL, where anthropogenic emissions play a significant role. Noteworthy is that the $RMSE_s$ values are in general higher for reanalysis products compared to STILT simulations (Fig. 8), resulting in average values of 9.5 ppm, 10.4 ppm, 7.4 ppm and 11.1 ppm for NTL, SDN, NGP, and MHL respectively. This indicates the advantage of using the STILT model over coarse-resolution models in utilizing these observations.

## 7    Conclusions

This study examines the potential of a high-resolution WRF-STILT modelling framework to simulate observed $CO_2$ variability over India. Further, we investigate the usability of these observations in inverse modelling when high-resolution models are used. Observations exhibit strong variability at seasonal (7.5-9.2 ppm) and intra-seasonal scales (2-12 ppm). STILT shows reasonable skill in representing the observed $CO_2$ variability in these stations, though the model could not sufficiently capture
every fine-scale observed variation. STILT simulations agree better with the observed seasonal and diurnal variations than the global reanalysis products. Among the reanalysis products, EGG4 products showed reasonable skill in predicting $CO_2$ variability over India.

STILT captures the seasonal variability associated with biospheric productivity in response to the availability of monsoon rainfall. But the model needs to account for small-scale flux variations like biomass burning to represent MHL observations.
Similarly, both STILT and global models did not capture the sharp reduction $CO_2$ concentration during August at the NTL station. This sharp reduction in $CO_2$ concentration is due to the increased biospheric uptake by the crops over the IGP region. The biospheric model could not represent this strong uptake, though the model reasonably well captured the seasonal variations in the site. More biospheric flux observations over the region may enable us to further improve our vegetation models by calibrating model parameters for different ecosystems.

The contribution of anthropogenic and biospheric components to the $CO_2$ concentration varies with the station and season. NTL shows strong $CO_2$ variability associated with biospheric fluxes. Over MHL, SDN, and NGP, anthropogenic flux variability in $CO_2$ concentration is dominant due to their proximity to cities. The impact of emission flux uncertainty in $CO_2$ variations is significant. $CO_{2ant}$ showed significant differences (up to 5 ppm) in their mean values and variability (up to 8 ppm) related to the choice of the emission inventory in the STILT model. The uncertainties in emission fluxes and their impact on $CO_2$ variations
indicate the importance of improving the inventories and their proper representation in the inverse modelling.

Our results show that observations from all these four stations can be utilised in the carbon data assimilation system with additional improvements in the modelling system. Except for NTL, the observations used in the study are modulated by influences from local fluxes in addition to background variations. Hence, most of these observations are suitable for constraining carbon fluxes at local-to-urban scales. NTL observations can be used in the regional carbon estimations as the observations
showed significant influences from regional fluxes. Given the availability of high-resolution fluxes, we demonstrate that the



STILT simulations can reasonably simulate the $CO_2$ variability over India. The availability of additional high-frequency observations representing the regional $CO_2$ variability over India, comparable to the World Meteorological Organization standards (https://gml.noaa.gov/ccl/co2_scale.html, last access: 12 June 2023) is necessary for improving our carbon estimates at scales relevant to policymaking.

*Code and data availability.*

The source code for WRF model version 3.9.1.1 that we used for the simulations of the meteorological fields is available from https://www2.mmm.ucar.edu/wrf/users/download/get_source.html (last access: 20 January 2022). STILT model source codes are available from https://stilt-model.org/ (last access: 15 March 2022). The EGG4 reanalysis products are availed from https://ads.atmosphere.copernicus.eu/cdsapp#!/dataset/cams-global-ghg-reanalysis-egg4?tab=form (last access: 10 February

2023). The CarboScope products used in this study are obtained from https://www.bgc-jena.mpg.de/CarboScope/ (last access: 20 July 2020). The CarbonTracker data is accessed from https://doi.org/10.25925/20201008 (last access: 22 October 2022). The EDGAR inventory data used in this study is from https://edgar.jrc.ec.europa.eu/dataset_ghg70 (last access: 19 January 2023). The ODIAC data is acquired from https://doi.org/10.17595/20170411.001 (last access: 23 February 2023). The ICOS data is obtained from https://hdl.handle.net/11676/-XUdi3MSHmJxSVBKmPmrTBOn (last access: 9 March 2022). GFAS data

is downloaded from https://ads.atmosphere.copernicus.eu/cdsapp#!/dataset/cams-global-fire-emissions-gfas?tab=form(last access: 24 November 2021). The Fire Inventory from NCAR version 2.5 is accessed from https://rda.ucar.edu/datasets/ds312.9/dataaccess/ (last access: 3 July 2023). The Nanital $CO_2$ observations are obtained from https://doi.org/10.17595/20220301.001 (last access: 10 December 2022), and Shadnagar and Nagpur observations are downloaded from https://bhuvan-app3.nrsc.gov.in/data/download/index.php (last access: 12 December 2022). Additional materials for the manuscript are available from

https://zenodo.org/record/8143361 (last access: 13 July 2023, Thilakan and Pillai (2023)).

*Author contributions.*

DP conceptualized, and VT and DP designed the study. VT developed the analysis methods, and VT and JS conducted the model simulations. HH, VS, YT and MN made the $CO_2$ measurements. VT conducted the analyses and wrote the first draft of the manuscript. MD performed the satellite data acquisition and analysis. DP and CG supported the interpretation of the

results. All authors discussed the results and provided feedback on the manuscript.

*Competing interests.*

At least one of the (co-)authors is a member of the editorial board of Atmospheric Chemistry and Physics.



*Acknowledgements.* This study has been supported by funding from the Max Planck Society allocated to the Max Planck Partner Group at IISERB. Vishnu Thilakan acknowledges the Ministry of Education (MoE) for his PhD funding. We acknowledge the support of IISERB's high-performance computing facility for STILT model simulations and data analysis. The WRF simulations were done on the high-performance cluster system (Levante) of the German Climate Computing Center (DKRZ). We acknowledge the IISER Mohali Atmospheric Chemistry Facility funded by the Ministry of Education, India, for the data and thank current and previous members of the Atmospheric Chemistry and Emissions and Aerosol Research groups for technical assistance. The Nainital measurements were supported by the Environment Research and Technology Development Fund (JPMEERF20182002 and JPMEERF21S20802) of the Environmental Restoration and Conservation Agency of Japan, ARIES and ISRO-ATCTM project. We are thankful to Shohei Nomura, Toshinobu Machida, Motoki Sasakawa, Hitoshi Mukai and Deepak Chausali for making possible the Nainital measurements. We thank the Climate and Atmospheric Processes of ISRO-Geosphere Biosphere Programme (CAP-IGBP) for providing observations from Shadnagar and Nagpur, as well as Mahesh Pathakoti for technical support.



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
