# Peer review of "Potential of using CO2 observations over India in regional carbon budget estimation by improving the modelling system"

_EGUsphere, 2023_

## Author Comment (AC1)

**Authors' Response to Referees:**

We greatly appreciate both referees for reviewing our manuscript and providing helpful comments and suggestions to improve our manuscript. We have addressed all the comments, suggestions and concerns raised by the referees and incorporated the associated modifications in the manuscript. By doing so, we believe that the manuscript builds on clarity and readability. Thank you.

Our responses and modifications in response to the reviewers' comments are listed below: the reviewers' comments are given in black regular font, our responses (All Authors) are given in blue regular font, and the changes in the revised version are given in *blue italic font.*
* * *
**Response to Referee 1 (R01) comments**

**R01**: In this manuscript, Thilakan et al. investigated the capability of a high-resolution Lagrangian chemical transport model to simulate spatiotemporal pattern of atmospheric $CO_2$ mixing ratios in India. The primary goal of this effort was about to assess the potential of $CO_2$ atmospheric mixing ratio measurement data in the regional and global inverse modeling systems to optimize $CO_2$ fluxes.

Their approach involved the use of WRF-STILT modeling system to simulate $CO_2$ concentrations at four sites in India. First, they described the $CO_2$ mixing ratio variability in four different $CO_2$ ground monitoring stations. Then they evaluated model simulations and three global model outputs against $CO_2$ observations from these four sites. They found an outperformance of WRF-STILT compared to three global models, in representing the spatiotemporal variability of observed $CO_2$. Finally, authors investigated the factors leading to model-observation differences. They conclude that regional and global data assimilation models can benefit from $CO_2$ measurement data if the models are improved in certain aspects.

This is a timely effort examining the potential of modeling systems to simulate $CO_2$ concentrations and infer fluxes driving those atmospheric mole fractions. $CO_2$ flux estimation in India is in its developing stages, as more measurement data from monitoring stations are now available (still sparse). Assessing the potential of $CO_2$ measurement data in the regional and global inverse modeling systems to accurately estimate $CO_2$ fluxes is a great effort.

Therefore, this paper is within the scope of Atmospheric Chemistry and Physics (ACP) journal and investigates an important research question. The high-resolution model

simulations were conducted, evaluated with observations, and model-data mismatch assessed. This manuscript can definitely be accepted to ACP, after addressing the below mentioned comments.

**All Authors (AA):** We greatly acknowledge Dr. Sajeev Philip for his careful review and appreciation.

We have addressed all his comments/suggestions/concerns, which are detailed below.

**Major comments**

**R01:** 1. Assessing the potential of observations in data assimilation models: The primary goal in this manuscript (and the title of the manuscript) was about to investigate the potential of $CO_2$ atmospheric mixing ratio measurements in the regional and global inverse modeling systems to optimize $CO_2$ Unfortunately, authors could not demonstrate this aspect in a robust manner. Authors found that: 1) the STILT model simulations outperformed global reanalysis products, 2) the STILT model still require some improvements in representing emission fluxes accurately, 3) and the contribution of systematic RMSE for STILT simulations is only a certain % of the overall error for certain seasons/stations. However, it is unclear how the potential of in situ observations in inverse models was assessed.

**AA:** Thank you for the comment. In this study, we assessed the potential of in-situ $CO_2$ observations in the regional inverse modelling of carbon fluxes by examining the variability in observations, the forward transport model's capability in capturing those variations, and characterising potential error structures for the carbon data assimilation. For constraining regional fluxes, highly variable measurements having a strong influence from local flux variations may not be suitable because of probable low regional representativeness. Hence, a part of our analysis dealt with understanding the causes of these observed measurement variations and their representativeness. The above variability analysis is required before utilizing mixing ratio observations to the carbon assimilation system. For assimilating regionally representative measurements from inland (this is the case for all measurements used in this study), a further requirement is a highly resolving forward model capable of deducing important information from the measured signals. Having analysed the observed variability, a logically imperative task is to implement a high-resolution modelling system that can reproduce these observed variabilities. Besides the need for accurate and continuous $CO_2$ observations, a carbon flux assimilation system heavily relies on how accurately the forward transport models simulate the fine-scale transport-related variations. Forward transport modelling errors need to be quantified and taken adequately into account in the inverse optimisation schemes to assign proper weights to the observations and determine the confidence level of the estimates. Hence, the contribution of this study includes a high-resolution

WRF-STILT model for the potential use of observations from India and an estimate of forward modelling errors for assessing how effectively observations can be utilised in future inverse optimisation systems.

Please also see our response to the next R01 comments (Point no. 2)

We have modified the manuscript for clarity as follows:

L73-76: "*This study focuses on assessing the potential of four available observations over India to be employed in future high-resolution Inverse modelling frameworks to optimise regional $CO_2$ fluxes. We analyse the variability and representativeness of $CO_2$ observations from each station. Observations with high variability, often due to the influence of local flux variations, may not be suitable for regional flux optimisations but can provide important information about local emission sources.*"

L506-510: "*The $CO_2$ variability over an observation site is influenced by the flux variability over its footprint region (see Figs. S1-S4). In the context of the inverse modelling that optimises $CO_2$ flux components (such as biospheric, anthropogenic, or both), it is important to ensure considerable information gain of relevant components when observations from a particular site are utilised. So here, we investigate the relative contribution of different components to the total $CO_2$ concentration from each observation site.*"

L555-557: "*An accurate estimation of $CO_2$ fluxes through inverse modelling demands proper accountability of $CO_2$ variability by the forward model employed. The disagreements between the observations and simulations largely arise from an inadequate representation of mesoscale transport and local flux influences.*"

L576-579: "*The fact that the majority of the uncertainty in STILT simulations over Nainital, Shadnagar and Nagpur contributed by unsystematic components shows the ability of the STILT model to represent the $CO_2$ variability there. Hence, these observations can be utilised in inverse optimisation with the help of high-resolution simulations from STILT.*"

L609-614: "*In other words, our study demonstrates a possible way of utilising observations from the Indian sub-continent that may potentially improve the global assimilation system estimates by increasing the degrees of freedom. Simultaneously, the availability of additional high-frequency observations representing the regional $CO_2$ variability over India, comparable to the World Meteorological Organization standards (https://gml.noaa.gov/ccl/co2_scale.html, last access: 12 June 2023) is necessary for improving the carbon estimates over India at scales relevant to policymaking.*"

**R01**: 2. Applicability to Eulerian models and global model: The authors tried to investigate the potential of observations in the WRF-STILT model (see major comment #1). The last statement of the abstract: "…the global carbon data assimilation system can thus benefit…". Authors should explain clearly how the results from this manuscript is applicable for regional/global Eulerian transport models (forward and inverse).

**AA:** Thank you for the comment. We added some statements in the manuscript to explain this better.

None of the current generation global carbon estimation systems utilises $CO_2$ measurements from India, leaving a substantial area less constrained with observations. The above choice is made mostly for two reasons: 1) the unavailability of continuous observations over the sub-continent and 2) the inability of forward models to represent the observed variations, mainly due to unresolved fine-scale transport. The model-observation mismatches due to unresolved variability (e.g. forward transport modelling error) can lead to significant uncertainties in the flux estimations, making observations less useful. Here, we investigate how a high-resolution model, as demonstrated in this study, can minimise the large forward modelling errors; thereby, the assimilation system can benefit the full potential of Indian observations. However, we also show that not all observations can be treated or filtered in a similar way, and care must be taken before they are assimilated. For example, stations such as Mohali, Shadnagar and Nagpur stations are strongly influenced by the local flux variations. Hence, information on local fluxes (urban and suburban scale flux estimations) can be deduced from these observations when a forward model reproduces the small-scale variabilities exhibited by the measurements. On the other hand, the Nainital observations are found to be regionally representative, for which the influence of local flux variations is less. Overall, our study demonstrates a possible way of utilising observations from the Indian sub-continent that can potentially improve the global assimilation system estimates by increasing the degrees of freedom.

609-611: *"In other words, our study demonstrates a possible way of utilising observations from the Indian sub-continent that may potentially improve the global assimilation system estimates by increasing the degrees of freedom. "*

**R01**: 3. All emissions to be added in the base simulations: The impact of biospheric fluxes and biomass burning emissions were assessed at some point in the manuscript. If authors had incorporated the biomass burning fluxes and an additional biosphere model (in addition to VPRM) to the base-model simulations, some sub-sections in Sect. 6 could have been avoided. Also, we all know that these emissions and local transport should be well represented in any kind of models. Authors should at least mention the novel aspects of this analysis conducted here (the impact assessments).

**AA:** We respectfully disagree on the irrelevance of sub-sections here. Section 6 is dedicated to a discussion critically examining how well a high-resolution modelling system can optimally deduce underlying fluxes at different spatial and temporal scales, given the potential of observations. For inverse modelling, it is crucial to understand the information content of the data to be assimilated. By doing these impact analyses, we also highlight the information on different flux components confined in the measured variability in addition to the extent of modelling errors. Also, we assess the extent of modelling errors due to misrepresenting prior fluxes (e.g. biomass burning), which is important for designing inverse optimization (e.g. defining error structure).

For instance, Mohali observations, depending on the season, show a considerable influence on biomass burning. These influences must be quantified and accounted for when utilising Mohali observations for urban or semi-urban scale flux estimates. Sect. 6.1 deals with these assessments and discussions. To examine the level of the model's disagreement with observations in Mohali due to prior fluxes, we used two different emission inventories: GFAS and FINNv2.5 (see Sect. 6.1). Simulations show a significant underestimation of the biomass burning-related enhancement (~15 times) over this region, irrespective of inventories. This indicates the limitations of the current emission inventories in representing agricultural residue burning over this region. On the other hand, the impact analyses showed a considerable potential of Nainital observations to optimise biospheric fluxes (via inverse modelling), which is discussed in Sect. 6.5. Note that assessing the sensitivities of different prior flux models to forward simulations is not part of this study's goal. Rather, we are interested in how one can potentially utilise these in-situ observations for informing underlying flux patterns.

However, we agree that some parts of the discussion need a better explanation. We have revised the manuscript as follows for clarity:

L431-442: "*In the previous sections, we have seen that the STILT model has improved capabilities in simulating these fine-scale variabilities. Here, we critically examine how well our modelling system can utilise observations from India to deduce optimal information on underlying fluxes at different spatial and temporal scales. The major implications of our results are discussed here, with the interest of further improving the carbon data assimilation over India.*

*We begin this section by exploring the shortcomings that need to be addressed to use potential $CO_2$ observations from India for inverse optimization (see Sect(s) 6.1 - 6.3). This is because three of the four observation sites used in this study (Mohali, Shadnagar and Nagpur) are situated near cities and are characterised by large intra-seasonal variability. Observations from all these four sites show strong seasonal variations in $CO_2$ concentrations (see Sect. 5.1). Along with the seasonal variations,*

*these observations (except Nainital) are also characterised by strong small-scale variability associated with local flux variations. It is thus challenging for coarse-resolution models to utilise them for inverse optimization. Thus, an account of contributions from different sources to the total observed $CO_2$ is discussed in Sect. 6.4, providing useful information about the underlying processes that these observations may carry.*"

L450-454: "*The potential of using Nainital observations via high-resolution inverse modelling to improve the ecosystem uptake and release is further elucidated in Sect. 6.5.*

*Using the systematic and unsystematic error components in model-data disagreements, Sect. 6.6 discusses the extent to which our model can utilise the full potential of the observations over India, thereby assessing the potential of observations to increase the confidence levels of the derived fluxes.*"

L473-481: "*Both of these data have a horizontal resolution of 0.1 × 0.1 and a temporal resolution of one day. The biomass emission fluxes over the footprint region of Mohali closely follow the fire count data (see Fig. S11a). GFAS emission fluxes over the Mohali footprint region are much less than FINNv2.5 (Fig. S11a). STILT simulations using FINN (STILT-FINN) indicate some influence from biomass burning with a time-lead with the $CO_2$ observations (see Fig. S11b). However, STILT simulations using GFAS (STILT-GFAS) could not represent the $CO_2$ contributions from biomass burning (Fig. S11b). The magnitude of $CO_2$ enhancement due to the biomass burning from STILT simulations is 3 ppm, which is ~15 times less than the magnitude of the emission enhancement observed in the $CO_2$ observations (see Fig. S11b). This suggests that the emission inventories need to be improved further to accurately simulate the $CO_2$ variability due to biomass burning in the region.*"

**R01**: 4. Novelty aspect to be added to the introduction section: The novel aspect of this study should be added to the introduction section with detailed citations of the relevant papers.

**AA:** Done, thank you. The manuscript text is revised as follows:

L68-78: "*There are increasing efforts in recent years to constrain the regional $CO_2$ fluxes over India via inverse modelling frameworks (Halder et al., 2022; Philip et al., 2022; Sijikumar et al., 2023). However, when assimilating regional measurements in the inverse optimisation framework, it is crucial to investigate how effectively the forward modelling system reproduces the observed variations associated with fine-scale transport and local influences at various time scales. This is because considerable*

*model-data mismatches due to transport errors can lead to large uncertainties in the estimated fluxes.*

*This study focuses on assessing the potential of four available observations over India to be employed in future high-resolution Inverse modelling frameworks to optimise regional $CO_2$ fluxes. We analyse the variability and representativeness of $CO_2$ observations from each station. Observations with high variability, often due to the influence of local flux variations, may not be suitable for regional flux optimisations but can provide important information about local emission sources. Further, we examine the capability of a high-resolution modelling framework based on the Lagrangian particle dispersion model (LPDM) to simulate $CO_2$ variability over these observation sites.*"

**R01**: 5. Need well-defined analysis approach: From the major comments #1 to #3, it is clear that a well-defined analysis approach should have been developed and described well in the start of Section 2 itself.

**AA:** Our entire methodology includes high-resolution modelling, data (observations and reanalysis products), and model skill assessments. We do think that a brief description of each of them is required for the ACP readership. Hence, we adapted to the following logical arrangements: Sect. 2 describes the modelling framework; Sect. 3 provides the details of $CO_2$ measurements used in this study; Sect. 4 details the analysis approach used for assessing the model skill in capturing observed variability.

The organisation of subsequent sections is mentioned just before Sect. 2 for improved readability.

**R01**: 6. Selection of sites and models: Better if authors mention clearly why certain monitoring stations (four) were selected. Just due to data availability? Also, better if they compared their model simulations to WRF-GHG model simulations (in addition to global products) from the same authors (Thilakan et al., 2022).

**AA:** There are very limited sites in India having systematic observations and therefore the selection of the observation sites was solely based on the data availability.

Please note that the WRF-GHG simulations from Thilakan et al., 2022 do not cover the full seasonal cycle for a direct comparison, so we do not include them in the evaluation. At the monthly scale, we see that both WRF-STILT and WRF-GHG have similar performance. We have modified the manuscript as follows:

L183-185: "*We used atmospheric $CO_2$ observations for 2017 from four measurement sites, located at Mohali, Nainital, Shadnagar and Nagpur (see Fig. 1) to assess their temporal variability. The sites were chosen based on the availability of their long-term measurements to the research team.*"

**R01**: 7. Presentation of results through summary figures: The manuscript comes with 9 (main manuscript) + 14 (supporting document) + 15 (additional materials: https://zenodo.org/record/8143361) = 38 figures! Most of the sentences in results and discussions sections are going back and forth between different figures, which makes it difficult to read and understand. Authors should be able to come up with <10 simple summary figures in the main manuscript to represent their results. For example, the subplots of the figures 2, 3, 5, 6, and 7 can be represented in a single plot with observation bar + 9 bars. Other options can be explored.

**AA:** Thank you for allowing us to refine the figures. We followed your suggestion and have modified the figures with panels. In the revised version, we have only five main figures.

**R01**: 8. Need clarity in conclusions: A major drawback of this manuscript is the loose conclusions in each and every section in Sect. 5 and 6. Authors first describe some figures and then make some vague general statements at the end of the paragraphs (see minor comments for some examples).

**AA:** Revised. Please see the revised manuscript (Sect(s). 6 and 7) and responses to the minor comments.

**R01**: 9. Presentation of results and discussions in the manuscript: Authors should consider revising/re-organizing the results (Sect. 5) and discussion (Sect. 6) sections to convey novel findings of this manuscript clearly. Also, abstract and conclusion sections need to be revised (see minor comments).

**AA:** Thanks, we have revised and reorganised Sect(s) 5 and 6 for better readability. We have also revised the abstract and conclusions. Please see the major changes below

L9-15: "*The seasonal $CO_2$ concentration variability in Mohali, associated with crop residue burning, is largely underestimated by the models. WRF-STILT captures the seasonal biospheric variability over Nainital better than the global models but underestimates the strength of the $CO_2$ uptake by crops. …… Our study highlights the possibility of using the $CO_2$ observations from these Indian stations for deducing carbon flux information at regional (Nainital) and sub-urban to urban (Mohali, Shadnagar and Nagpur) scales with the help of a high-resolution model.*"

L431-442: "*In the previous sections, we have seen that the STILT model has improved capabilities in simulating these fine-scale variabilities. Here, we critically examine how well our modelling system can utilise observations from India to deduce optimal information on underlying fluxes at different spatial and temporal scales. The major implications of our results are discussed here, with the interest of further improving the carbon data assimilation over India.*"

*We begin this section by exploring the shortcomings that need to be addressed to use potential $CO_2$ observations from India for inverse optimization (see Sect(s) 6.1 - 6.3). This is because three of the four observation sites used in this study (Mohali, Shadnagar and Nagpur) are situated near cities and are characterised by large intra-seasonal variability. Observations from all these four sites show strong seasonal variations in $CO_2$ concentrations (see Sect. 5.1). Along with the seasonal variations, these observations (except Nainital) are also characterised by strong small-scale variability associated with local flux variations. It is thus challenging for coarse-resolution models to utilise them for inverse optimization. Thus, an account of contributions from different sources to the total observed $CO_2$ is discussed in Sect. 6.4, providing useful information about the underlying processes that these observations may carry."*

L450-454: *"The potential of using Nainital observations via high-resolution inverse modelling to improve the ecosystem uptake and release is further elucidated in Sect. 6.5.*

*Using the systematic and unsystematic error components in model-data disagreements, Sect. 6.6 discusses the extent to which our model can utilize the full potential of the observations over India, thereby assessing the potential of observations to increase the confidence levels of the derived fluxes."*

L586-588: *"To utilise these observations in inverse optimisations, the models need to address model-observation mismatches arising from fine-scale trans- port and local flux influences."*

L592-595: *"Further, we explored the limitations of the STILT modelling system in representing the variability, although the model captures the intra-seasonal variabilities much better than the global models. However, the model must account for small-scale flux variations like biomass burning to represent Mohali observations. A considerable portion of these discrepancies can be minimized by improving the prior emission flux distribution at high spatial and temporal scales"*

L597-599: *"An improved estimate of biospheric $CO_2$ uptake and release can be achieved by utilising these $CO_2$ observations to optimise the fluxes through carbon data assimilation or by using flux observations from different ecosystems to calibrate the model parameters over India (e.g., Ravi et al., 2023)."*

L608-611: *"Given the availability of high-resolution fluxes and better representation of the fine-scale transport, we demonstrate that the STILT can reasonably simulate the $CO_2$ variability over India. In other words, our study demonstrates a possible way of*

*utilising observations from the Indian sub-continent that may potentially improve the global assimilation sys- tem estimates by increasing the degrees of freedom.*"

**Minor comments**

**R01**:

Line 3: Better to use: "high-resolution Lagrangian chemical transport model"

Line 6: Better use: "Our modeling framework"

**AA:** Done

**R01**: Line 8: How about sub-seasonal?

**AA:** The performance of STILT in simulating sub-seasonal variability is also better than the current generation models. The details are added in the revised version (Sect(s). 5.2 and 5.3).

L335-336: "*STILT simulations show an RMSE of 8-9 ppm with the observed intra-seasonal variability in Mohali. The intra-seasonal variability is derived by removing the monthly mean values from the $CO_2$ concentration.*"

L363-364: "*The estimated mismatches for intra-seasonal variability between observation and STILT simulations at Nainital is ~ 4 ppm (based on RMSE)*"

L375-376: "*The estimated intra-seasonal variability shows an RMSE of 4.3 - 6.0 ppm with STILT simulations over Shadnagar.*"

L380-381: "*STILT simulations show an RMSE of 6.5 - 11.5 ppm with the estimated intra-seasonal variations over Nagpur.*"

L395: "*Global models show an RMSE of ~ 10 ppm with intra-seasonal variability of observations over Mohali.*"

L411-412: "*A model-observation mismatch of ~ 5 ppm (RMSE) is observed in the intra-seasonal variability estimated over Nainital.*"

L420: "*Global models show an RMSE of up to 7 ppm with intra-seasonal variability of observations over Shadnagar.*"

L428-429: "*The estimated intra-seasonal variability between global models and observations over Nagpur shows an RMSE of up to 10 ppm.*"

**R01**: Line 8: Diurnal only for one site? If so, better specify that.

Line 8: Be very specific: "current general global models".

**AA:** We have modified the abstract as follows:

L7-9: *"...performs better in simulating seasonal ($R^2$ = 0.50 to 0.96) and diurnal ($R^2$ = 0.96) variability (for Mohali station) of observed $CO_2$ than the current generation global models (CarboScope, CarbonTracker and ECMWF-EGG4)."*

**R01**: Line 9: Representation of all processes and emission categories needs refinement irrespective of the site location. I would say its an empty and loose sentence.

**AA:** We agree that the adequate representation of all processes (including emissions) is required in every models; however, not all site measurements have similar requirements depending on factors such as location, season, and time of the day. For example, the contribution of local flux changes like biomass burning on $CO_2$ variability is not uniform for all stations.

We have revised the manuscript as follows:

L9-12: "*The seasonal $CO_2$ concentration variability in Mohali, associated with crop residue burning, is largely underestimated by the models. WRF-STILT captures the seasonal biospheric variability over Nainital better than the global models but underestimates the strength of the $CO_2$ uptake by crops.*"

**R01**: Lines 9:14: Abstract should be having take-home messages from your article. All these lines are general statements, without any quantitative info.

Line 14: Better rewrite this sentence for clarity: "By implementing a model, our results emphasize…".

Line 16: Observed variability of …

**AA:** Done. We have modified the abstract as follows for better clarity:

L9-17: "*…The seasonal $CO_2$ concentration variability in Mohali, associated with crop residue burning, is largely underestimated by the models. WRF-STILT captures the seasonal biospheric variability over Nainital better than the global models but underestimates the strength of the $CO_2$ uptake by crops. The choice of emission inventory in the modelling framework alone leads to significant biases in simulations (5 to 10 ppm), endorsing the need for accounting emission fluxes, especially for non-background sites. Our study highlights the possibility of using the $CO_2$ observations*

*from these Indian stations for deducing carbon flux information at regional (Nainital) and sub-urban to urban (Mohali, Shadnagar and Nagpur) scales with the help of a high-resolution model. On accounting for observed variability of $CO_2$, the global carbon data assimilation system can benefit from the measurements from the Indian subcontinent."*

**R01**: Line 21: greenhouse gas

**AA:** Done

**R01**: Line 29: Change: "suffer from"

**AA:** We have modified the manuscript as follows:

L29: *"...these estimates are often characterised by large errors…"*

**R01**: Lines 32 and 35: Better to discuss about $CO_2$ rather than carbon fluxes

Line 36: Ganesan et al. deals with methane. Better to discuss about $CO_2$.

**AA:** In that paragraph, we describe the importance of carbon estimation through inverse optimization. We attempt to bring in major studies in this area at global and regional levels. Hence, we wish to keep this citation, not restricting the discussion only to $CO_2$. We have modified the manuscript as follows for clarity.

L34-37: *"There have been a few recent inverse-based attempts to estimate the carbon fluxes over the South Asian region using in-situ and satellite observations (e.g., Patra et al., 2013; Thompson et al., 2016; Ganesan et al., 2017; Philip et al., 2022; SijiKumar et al., 2023); however, these studies are limited by the general paucity of observational data with sufficient temporal and spatial coverage over the region."*

**R01**: Line 34: Restrict # of citations. Add those relevant to India.

Lines 34-36: Better mention that these models require observations to constrain fluxes. That way, this sentence connects with the next sentence (Lines 36-38).

**AA:** Thank you. We have modified the manuscript as follows:

L34-37: *"There have been a few recent inverse-based attempts to estimate the carbon fluxes over the South Asian region using in-situ and satellite observations (e.g., Patra et al., 2013; Thompson et al., 2016; Ganesan et al., 2017; Philip et al., 2022; SijiKumar et al., 2023); however, these studies are limited by the general paucity of observational data with sufficient temporal and spatial coverage over the region."*

**R01**: Line 39-40: Rewrite for clarity: "detect surface variations in addition to data gaps".

**AA:** We have modified the manuscript as follows:

L38-40: *"In-situ observations are essential for the tropics because satellite observations representing the entire atmospheric column cannot always detect signatures from small-scale surface flux variations. Moreover, one may expect significant data gaps in satellite measurements, depending on the season, due to clouds and moist convection."*

**R01**: Line 41: Are there any dedicated network? Or it's just individual observation stations? Also, there could be some more papers describing recent in situ observations.

**AA:** We have modified the manuscript as follows:

L40-42: *"Recently, more greenhouse gases (GHG) monitoring stations have been set up over India by different research initiatives (e.g., Tiwari et al., 2014; Lin et al., 2015; Mahesh et al., 2015; Chandra et al., 2016; Jain et al., 2021; Nomura et al., 2021; SijiKumar et al., 2023)."*

**R01**: Line 48: Not sure these constraints are the reason not to use Indian observations in global models (data availability…).

**AA:** We have modified the manuscript as follows:

L50-51: *"Due to the unavailability of long-term consistent observations representing the regional fluxes, none of the current generation global carbon assimilation systems utilises CO$_2$ observations from the Indian region"*

**R01**: Lines 48-50 and 55: There should be at least one paper (Sijikumar et al. 2023): https://doi.org/10.1016/j.atmosenv.2023.119868.

**AA:** We have added the following sentence to the manuscript:

L68-69: *"There are increasing efforts in recent years to constrain the regional CO$_2$ fluxes over India via inverse modelling frameworks (Halder et al., 2022; Philip et al., 2022; Sijikumar et al., 2023)."*

**R01**: Lines 53-54: Better to avoid "associated with… transport and flux".

**AA:** Done

**R01**: Line 59: Not sure if Thompson et al. found convection responsible for the difficulty in simulating CO$_2$ over tropics.

**AA:** Thompson et al. (2014) studied the $N_2O$ emission estimates with the help of multiple atmospheric inversion frameworks. In that study, they point out that one of the primary reasons for large uncertainty in estimates is the inability of transport models to simulate tropical convection properly. We assume that this will be the same case for other atmospheric tracers as well. We have modified the manuscript as follows for clarity:

L59-61: *"Similarly, local and large-scale convections play a major role in distributing atmospheric tracer concentrations (Gerbig et al., 2003) vertically, which is difficult to simulate in tropical regions (Thompson et al., 2014)."*

**R01**: Line 62: Not just over India.

**AA:** We have modified the manuscript as follows for more clarity:

L62-64 : "*Thilakan et al. (2022) showed that considerable representation errors exist when we use coarse-resolution transport models for inverse optimisation over India, and the representation error tends to decrease when we increase the horizontal resolution.*"

**R01**: Lines 65-66: It's not just about resolution. Modeling system should accurately represent all kinds of processes etc.

**AA:** We have modified the manuscript as follows:

L66-67: *"Hence, an adequate representation of the atmospheric $CO_2$ distribution over India relies on a modelling system that can operate at a high spatial and temporal resolution and has the ability to simulate all the essential underlying processes."*

**R01**: Lines 67-84: This paragraph should better fit in the material and methods section. Section 2 can start with the overall approach of this study. Here, in the introduction section, this paragraph can be described in 2-3 sentences.

**AA:** Done. Please see the revised manuscript.

**R01**: Line 77: Correct this: "indented"

**AA:** Done

**R01**: Line 115: Briefly mention how PBLH is calculated, in one or two sentences.

**AA:** We have added the following sentence to the manuscript.

L126-127: *"STILT compute the PBL height with the help of a modified Richardson number as described in Vogelezang and Holtslag (1996)."*

**R01**: Line 116: Which approach? Approach to calculate PBLH?

**AA:** We have modified the manuscript as follows:

L127-128: *"The relation between F(x, y, t) and S(X, t) is summarised in Eq. (3) as follows:"*

**R01**: Line 132: Add some more info: "which found it promising".

**AA:** The manuscript is modified as follows:

L142-145: *"The evaluation of the WRF model simulations over India shows a good agreement with observations (e.g., Hariprasad et al., 2014; Boadh et al., 2016; Sivan et al., 2021; Mathew et al., 2023). Mathew et al. (2023) demonstrated that temperature, moisture and wind simulations by WRF had a good correlation ($R^2 > 0.75$) in comparison with observations."*

**R01**: Line 137: Which satellite product?

**AA:** We have modified the manuscript to include that information as follows:

L149-150: *"...WRF meteorological fields and MODIS (Moderate Resolution Imaging Spectroradiometer) data from Terra and Aqua satellites."*

**R01**: Lines 135-144: Three anthropogenic inventories were used. Why not different biosphere flux models?

**AA:** Rather than assessing the sensitivity of different biospheric flux models to the total $CO_2$ simulation, we are interested in investigating the potential of observations to provide more information about the flux distribution, for example, terrestrial biospheric fluxes.

**R01**: Line 141: Already mentioned that "derives…at receptor locations using Eq. (4)" in lines 132 and 138.

**AA:** Thank you. We have removed the above sentence from the manuscript.

**R01**: Lines 143-146 and 162-166: Better to have a Table with the details of different model experiments and data/inventory sources.

**AA:** We have added a table describing different model experiments. Please see the modified manuscript.

**R01**: Lines 155:158 and Eq. (5): Why not biomass burning fluxes? Other chemical sources of $CO_2$ (may not be a significant fraction)?

**AA:** We have limited the use of biomass burning fluxes to Mohali station since its influence is significant only at Mohali station during crop harvesting seasons.Please see Sect. 6.1

We have modified the manuscript as follows for more clarity:

L166-168: *"We do not include the contribution from oceanic fluxes, as its influence on these stations is very negligible (~0.001 ppm, Figure not shown). We incorporate the influence of biomass burning separately over Mohali during the biomass burning emission season (see Sect. 6.1 for a detailed discussion)."*

**R01**: Line 169: Why not refer to these four monitoring stations with names of the cities (e.g., Mohali) rather than short forms (e.g., MHL).

**AA:** We have modified the manuscript as per your suggestion. Please see the revised manuscript.

**R01**: Lines 180-181: Better mention those three cities, not just Chandigarh.

**AA:** Done.

**R01**: Lines 232-235: Rewrite for clarity. First level of CT for all stations. Then explain the rest.

**AA:** The manuscript is revised as follows:

L247-250: "*To compare with the observations, simulations from the first vertical level of the CarbonTracker are used. Model simulations at 1000 mb pressure level from CarboScope and EGG4 are used to compare with the observations at Mohali, Shadnagar and Nagpur. Since Nainital is a mountain site situated at ~ 800 mb height, we compared those observations with CarboScope and EGG4 products at 800 mb vertical level.*

**R01**: Section 4: Good metrics from Willmott, (1981) and Willmott et al. (2012)!

**AA:** Thank you.

**R01**: Line 266: Any evidence shown here?

**AA:** We have modified the manuscript as follows:

L292-293: *"A similar reduction in $CO_2$ variability can be seen at 09:00 LT during May-August (Fig. S7), most likely due to a well-established convective boundary layer with strong mixing. "*

**R01**: Lines 269-270: Better assess diurnal aspects in a separate section along with a figure. All analysis and main figures should be based on daytime values.

**AA:** In the revised manuscript, we have explained the diurnal variability in a separate paragraph. Also, we have moved Fig. 3 in the old version (Diurnal variability) to the supplementary files.

**R01**: Line 261: The seasonal aspects can be described in a separate para.

**AA:** Done

**R01**: Lines 276-277: Better if you cite relevant publications to support this statement.

**AA:** Done. We have modified the manuscript as follows:

L285-287: *"However, we find high $CO_2$ concentrations at Mohali during November, which can be attributed to the agricultural waste-burning activities prominent around this region at this time of the year (Deshpande et al., 2023). A detailed discussion on the influence of biomass burning on $CO_2$ concentration over Mohali is provided in Sec 6.1. "*

**R01**: Lines 277-279: Any reason why the $CO_2$ contribution from biomass burning was not included as part of the base simulations?

**AA:** We have limited the use of biomass burning fluxes to Mohali station since its influence is significant only at Mohali station during crop harvesting seasons.Please see Sect. 6.1

**R01**: Figure 2: In the figure caption, describe what these color bars represent. Error bars?

**AA:** Done, We have modified the caption as follows:

*"Figure 2. $CO_2$ monthly variations over different stations during 2017. Observed $CO_2$ variability is shown in comparison with STILT simulations and global reanalysis products. Box and whisker plot of observation in comparison with model simulations is shown. The box denotes the interquartile range, and the whiskers represent the points within 1.5 times the interquartile range from the lower and upper quartile. Additionally, mean values for the $CO_2$ concentration are provided as a black circle inside the box.*

*Daytime (11:00-16:00 local time) values of the Mohali observations and simulations are used.”*

**R01**: Figure 2: The subplots of the figure 2 (this is applicable for all figure 3 and 5 to 7) can be represented in a single plot with observation bar + 9 bars. Or explore any other figure options to represent your results in a clearly.

**AA:** We have modified the figures. Please see the revised manuscript.

**R01**: Section 5.1: Not sure if these aspects were already mentioned in publications from research groups who conducted the $CO_2$ measurements (e.g., Sreenivas et al 2016)?

**AA:** Nomura et al., 2021 (Nainital) and Sreenivas et al., 2016 (Shadnagar) examined the seasonal variations over these stations. However, since they used different study periods for the analysis. We have modified the manuscript as follows to include this information.

L298-299: *“Nomura et al. (2021) also reported a similar seasonal cycle over Nainital, with lower values during February-March and September.”*

L304-305: *“At Shadnagar, measurements show a seasonal $CO_2$ variability of 4.4 ppm, with two higher peaks during April (404.7 ppm) and October (405.6 ppm), while Sreenivas et al. (2016) reported only one seasonal peak during pre-monsoon.”*

**R01**: Section 5.2 and 5.3: Hard to read. Going back and forth between different figures. Better to have some summary figures to demonstrate that STILT model outperforms global reanalysis products (obviously). Authors should consider revising/re-organizing the results (Sect. 5) and discussion (Sect. 6) sections to convey novel findings of this manuscript clearly.

**AA:** We have modified the figures to summarise the results. We have also revised Sect.5 and Sect. 6 for better clarity. Please see the revised manuscript.

**R01**: Line 415: Revise: “Using the STILT model has improved the capabilities…”

**AA:** We modified the manuscript as follows:

L431-432: *“In the previous sections, we have seen that the STILT model has improved capabilities in simulating these fine-scale variabilities.“*

**R01**: Line 418: I doubt you are trying to improve data assimilation “approaches”.

**AA:** We modified the manuscript as follows:

L433-434: *"The major implications of our results are discussed here, with the interest of further improving the carbon data assimilation over India."*

**R01**: Lines 435-439: Better if you show a figure with the biomass emissions from those two inventories (in addition to STILT simulations with those emissions).

**AA:** We have added the variability of biomass burning emission over Mohali to Fig. S11. The manuscript is modified as follows:

L474-476: "*The biomass emission fluxes over the footprint region of Mohali closely follow the fire count data (see Fig. S11a). GFAS emission fluxes over the Mohali footprint region are comparatively much less than FINNv2.5 (Fig. S11a).*"

**R01**: Line 440: Not clear to me: "Along with mixing height issues…"?

**AA:** We modified the manuscript as follows:

L482-483: *"Besides the inaccurate estimation of mixing height, misrepresentation of emission fluxes, as seen here, can lead to significant errors…"*

**R01**: Lines 440-442: We all know that misrepresentation of emission fluxes can lead to errors in the simulated concentrations. If authors had incorporated the biomass burning fluxes to the base simulations, this entire sub-section 6.1 could have been avoided (?) I wonder what new aspect is critically examined here. Also, not specified the resolution of the flux inventories. Not sure how you came to this conclusion: "…shows the role of high-resolution biomass burning fluxes…".

**AA:** Due to the significant underestimation of biomass burning fluxes by inventories (please see the response to the Major comment 3), STILT can not represent the enhancement related to biomass burning adequately. We have modified the manuscript to address your comments as follows:

L473-474: "*Both of these data have a horizontal resolution of 0.1°× 0.1°and a temporal resolution of one day.*"

L483-484: "*This result shows the role of high-resolution fluxes that can account for small-scale events like agricultural waste burning in representing $CO_2$ variability at Mohali.*"

**R01**: Lines 466-468: Clearly mention: GPP in Fig. S14a and NDVI in Fig. S14b.

**AA:** We modified the manuscript as follows:

L545-547: *"Noticeably, the simulated $CO_2$ uptake component contribution from Gross Primary Production (GPP), see Fig. S12a) from crops shows a similar pattern as that of the Sentinel-2 derived NDVI (see Fig. S12b) in the influence region of Nainital."*

**R01**: Lines 474-476: This second part of the sentence is not clear: "…and by using…".

**AA:** We modified the manuscript as follows:

L597-599: *"An improved estimate of biospheric $CO_2$ uptake and release can be achieved by utilising these $CO_2$ observations to optimise the fluxes through carbon data assimilation or by using flux observations from different ecosystems to calibrate the model parameters over India (e.g., Ravi et al., 2023)."*

**R01**: Section 6.3: So far, the authors assessed the impact of biomass burning emissions and biospheric flux model on the variability of $CO_2$ concentration. Now, they discuss relative contribution of different components. There is no coherence. Biosphere component is already discussed in Section 6.2.

**AA**: Section 6 is revised (please see our response to comments above). In Sect. 6.5 (revised manuscript) we have pointed out the importance of crops determining the biospheric fluxes over India. In Sect. 6.4, we investigate the dominant emission components over each observation site to understand the information gain that can be achieved while using these measurements in inverse modelling. We have added the following sentences to the manuscript for clarity.

L506-510: *"The $CO_2$ variability over an observation site is influenced by the flux variability over its footprint region (see Figs. S1-S4). In the context of the inverse modelling that optimises $CO_2$ flux components (such as biospheric, anthropogenic, or both), it is important to ensure considerable information gain of relevant components when observations from a particular site are utilised. So here, we investigate the relative contribution of different components to the total $CO_2$ concentration from each observation site."*

**R01**: Lines 499-501: Very loose conclusions. We all know these aspects. I wonder what new aspect is critically examined here. This is applicable for these paragraphs in Section 6.

**AA:** Section 6 is revised. In this study, we quantify the influence of emission uncertainties on $CO_2$ mixing ratio simulations by evaluating them with the observations. Hence the referred statement is not qualitative driven, but linking to the quantification metioned in statements in the manuscript. Please see the revised manuscript (pp. 20, Lines:487-493).

**R01**: Section 6.6: Section 6 started with an aim to explore the shortcomings to be addressed for using Indian $CO_2$ observations in inverse models. In sections 6.1 to 6.5 author noted the inherent issues with their modeling system. Now, in section 6.6, they try to assess the potential of in situ observations in inverse model. Structuring of different sections is not in a coherent manner. Sect. 6.6 should be a stand-alone section.

**AA:** We have reorganized Section 6. Please see the revised manuscript.

**R01**: Lines 510-511: better not to start a new section of paragraph with some conclusions: "…from issues in representing…".Lines 513-514: The approach by the authors (to assess the potential of observation in models) is now somewhat clear. Seems like an important section/paragraph…

**AA:** We have added the following sentence to the beginning of the paragraph.

L555-556: *"An accurate estimation of $CO_2$ fluxes through inverse modelling demands a proper representation of $CO_2$ variability in its forward simulations. The disagreements…"*

**R01**: Line 515: Better start with describing MHL (top left in the figure).

**AA:** Done.

**R01**: Figure 8: Caption: describe in the order: Blue and black lines represent…dr and r.

**AA:** Done.

**R01**: Figure 8: Caption: Describe in the right way "(a) Mohali (b) Nainital (c) Shadnagar (d) Nagpur"

**AA:** We have modified the caption as follows:

Fig.4. *"....The panels represent (a) Mohali, (b) Nainital, (c) Shadnagar, and (d) Nagpur stations."*

**R01**: Lines 517-520: Without a figure in the main manuscript, it is difficult to understand.

**AA:** We have moved the figure to the supplementary file. Please see supplementary figure Fig. S13

**R01**: Line 522: An empty/loose sentence. Better revise.

**AA:** We have modified the manuscript as follows.

L576-579: "*The fact that the majority of the uncertainty in STILT simulations over Nainital, Shadnagar and Nagpur is due to unsystematic components shows the ability of these simulations to represent the $CO_2$ variability there. Hence, these observations can be utilised in inverse optimisation with the help of high-resolution simulations from STILT.*"

**R01**: Line 523: Beter starts this sentence like this: "Figure 8a shows that the Mohali model-data mismatch is more systematic…". Just a suggestion.

**AA:** Thank you. We have modified the manuscript as follows:

L561: *"Figure 4a shows that the Mohali model-observation mismatch is more systematic (66-86 %) in nature. High…"*

**R01**: Line 533-534: Obviously coarse resolution model fails in representing fine-scale features as compared to high-resolution models (with a more realistic representation of local influences).

**AA:** Yes, depending on the location and representativeness of the observations. When fine-scale transport and local flux variability dominate on influencing total $CO_2$ observations, the coarse resolution models tend to give considerable model-observation mismatches.

**R01**: Section 6.6: Not sure if the second objective of the article (to assess the potential of observational data in inverse modeling) is investigated in detail.

**AA:** Please see our response to Major comment 1.

**R01**: Line 540: Revise: "…capture every fine-scale…".

**AA:** We have modified the manuscript as follows:

L588-589: *"Our model shows reasonable skill in representing the observed $CO_2$ variability in these stations, though the model could not sufficiently capture all the observed fine-scale variations."*

**R01**: Lines 545-546: Better combine these two sentences.

**AA:** We have modified the manuscript as follows:

L595-597: "*Both STILT and global models did not capture the sharp reduction $CO_2$ concentration at Nainital during August, resulting from the increased biospheric uptake by the crops over the IGP region.*"

**R01**: Conclusion section: This section should be described in a coherent manner. Right now, it is just a collection of some select sentences from the results and discussion sections.

**AA:** We have modified the conclusion section for better clarity. Please see the revised manuscript.

**R01**: Line 555: Not just in inverse modeling. These aspects are important for any models, forward or in inverse modes.

**AA:** We have modified the manuscript as follows:

L600-601: *"The extent of uncertainties in emission fluxes and their impact on $CO_2$ variations indicate the importance of improving the inventories and their proper representation in atmospheric transport modelling and inverse estimations."*

**R01**: Line 556-557: Not sure how authors proved it.

**AA:** We have revised  as follows for clarity:

L604-613: "*Except for Nainital, the observations used in the study are modulated by influences from local fluxes in addition to background variations. Hence, most of these observations are suitable for constraining carbon fluxes at local-to-urban scales. Nainital observations can be used in the regional carbon estimations as the observations showed significant influences from regional fluxes. Given the availability of high-resolution fluxes and better representation of the fine-scale transport, we demonstrate that the STILT can reasonably simulate the $CO_2$ variability over India. In other words, our study demonstrates a possible way of utilising observations from the Indian sub-continent that may potentially improve the global assimilation system estimates by increasing the degrees of freedom. Simultaneously, the availability of additional high-frequency observations representing the regional $CO_2$ variability over India, comparable to the World Meteorological Organization standards (https://gml.noaa.gov/ccl/co2_scale.html, last access: 12 June 2023) is necessary for improving the carbon estimates over India at scales relevant to policymaking.*"

**R01**: Line 563: Revise: "…improving our carbon estimates…".

**AA:** We have modified the manuscript as follows:

L612-613: "...*is necessary for improving the carbon estimates over India at scales relevant to policymaking.*"

**Response to Referee 2 (R02) comments**

**R02**: Title: Potential of using $CO_2$ observations over India in regional carbon budget estimation by improving the modelling system by Thilakan et al.

The authors have conducted an analysis using a high-resolution transport model to assess the potential of $CO_2$ observations over India for the inverse estimation of regional carbon fluxes. They have also evaluated the performance of the region-specific models by comparing them to surface-based observations and using statistical metrics. This contribution is undoubtedly of interest to the scientific community, although it necessitates careful interpretation in the presentation of results. Nonetheless, there are certain important scientific aspects that have not addressed properly and therefore have provided specific comments and suggestions that necessitate clarification in the revised manuscript. Therefore, I recommend that it undergoes Major revision, incorporating all the suggestions and comments provided.

**AA:** We thank Referee 2 for reviewing our manuscript and providing helpful suggestions. We have addressed all concerns, comments, and suggestions by R02. Please see our responses to the comments below.

**Major Comments:**

**R02**: In the introduction, specifically on line 40, the authors emphasize the ongoing efforts on the surface-based $CO_2$ observations over India but do not reference previously published studies that have reported on the diurnal and seasonal cycle variability in the Indian region. For instance, studies such as Chandra et al. (2016) and Jain et al. (2021) conducted at Gadanki, among others, have explored this aspect. It would be beneficial to include references to existing literature on ground-based $CO_2$ studies in India.

**AA:** Thank you. We have revised the manuscript to incorporate these citations.

L40-42: "*Recently, more greenhouse gases (GHG) monitoring stations have been set up over India by different research initiatives (e.g., Tiwari et al., 2014; Lin et al., 2015; Mahesh et al., 2015; Chandra et al., 2016; Jain et al., 2021; Nomura et al., 2021; Sijikumar et al., 2023).*"

**R02**: Furthermore, there are ongoing modeling efforts in India aimed at representing $CO_2$ variability and fluxes using high-resolution models. For instance, studies by Halder et al. (2021) and Siji Kumar et al. (2023), among others, have contributed to this field. It

is essential to incorporate literature on these modeling efforts in order to enhance clarity and identify areas where the modeling framework may have gaps.

**AA:** Thank you. Revised the manuscript as follows**:**

L69-70: *"There are increasing efforts in recent years to constrain the regional $CO_2$ fluxes over India via inverse modelling frameworks (Halder et al., 2022; Philip et al., 2022; Sijikumar et al., 2023)."*

**R02**: One major concern in the design of the model simulations is the omission of the role played by biomass burning and oceanic fluxes. Although their contributions are minimal, I believe that the authors should also take these fluxes into account when simulating $CO_2$ concentrations at specific sites. For instance, Mohali, one of the sites, is heavily impacted by biomass/ crop residue burning. More details to be provided during the revision process.

**AA:** Depending on the season and location, there is a considerable contribution of biomass burning to the $CO_2$ variability. We have included a detailed discussion about this contribution in Sect. 6.1, and incorporated the biomass-burning component in Mohali simulations by using inventories such as GFAS and FINNv2.5. For other sites, there are zero or negligibly small contributions from biomass burning; hence, STILT simulated enhancements are unaffected.

The influences of ocean fluxes for all stations are insignificant due to their locations far from the oceanic region (please see Fig. R1). Hence, it is not necessary for the model configuration to include oceanic fluxes when representing these specific sites. We have modified the manuscript as follows for more clarity:

L166-168: *"We do not include the contribution from oceanic fluxes, as its influence on these stations is very negligible (~0.001 ppm, Figure not shown). We incorporate the influence of biomass burning separately over Mohali during the biomass burning emission season (see Sect. 6.1 for a detailed discussion)."*

**R02**: In Section 5.1, authors made an attempt to address the seasonal cycle of $CO_2$ observations and model simulations. I believe it would be more beneficial to focus on a discussion of the model's strengths and weaknesses in capturing seasonality, rather than solely presenting the comparison results. In some instances, the model appears to be significantly out of sync with the observations, and it would be valuable to provide explanations for these discrepancies.

**AA:** We have revised the manuscript to give more details on seasonal variability. Fig.3 (Seasonal variability of $CO_2$ observations and model simulations) is added to the main manuscript. Please see the revised manuscript. We have also discussed the strength and weakness of the model in capturing the seasonality. All these are detailed and revised in Sect(s). 5.2 and 5.3.

**R02**: Discussing the observations alone and later (section 5.2) section about the model comparison doesn't make much information. If authors want to make robustness in the model simulations, better to provide them together for a better clarity. It would better to reorganize these sections and if needed make one section to discuss both observational and model discussions together.

**AA:** We respectfully disagree. We attempted to adapt a logic flow in which we first quantify the variability in observations to assess their potential in informing underlying flux patterns. For instance, highly variable observations are often influenced by local flux variations in addition to transport mechanisms, which may not be suitable for regional flux estimations. Sect. 5.1 is hence designed for the above purpose. The next step is to investigate whether our modeling system is sufficient enough to utlise these observations. Given the considerable variability in the observations, global models are unable to resolve these and utilise them in the assimilation, necessitating the high-resolution models to be employed (see Sec. 5.2).  Considerable model-observation mismatch due to unresolved transport and local flux patterns can lead to significant posterior flux uncertainties. We have revised the manuscript to explain these better, as follows:

L73-78: "*This study focuses on assessing the potential of four available observations over India to be employed in future high-resolution Inverse modelling frameworks to optimise regional $CO_2$ fluxes. We analyse the variability and representativeness of $CO_2$ observations from each station. Observations with high variability, often due to the influence of local flux variations, may not be suitable for regional flux optimisations but can provide important information about local emission sources. Further, we examine the capability of a high-resolution modelling framework based on the Lagrangian particle dispersion model (LPDM) to simulate $CO_2$ variability over these observation sites.*"

L81-83: "*We assess the usability of these measurements in the inverse framework when utilising the high-resolution (e.g., WRF-STILT) modelling system to optimise carbon fluxes. We quantify the model uncertainties and compare them with some of the existing coarse-resolution models.*"

**R02**: In Figure 3, why did the authors choose to present three-hourly averages instead of reporting the hourly means of observations and model simulations, if the data for hourly means is available?

**AA:** We have configured STILT to generate 3-hourly simulations, consistent with all other stations. These simulations are found to be capable of representing the diurnal variations in observations. Further, 3-hourly averaging can minimise the random fluctuations, which is of less interest in this study. We are rather interested in the systematic model biases. Please note that most of the assimilation system uses temporally averaged data (e.g. daytime average). Nevertheless, we have provided here a comparison of diurnal averages calculated from 3-hourly and 1-hourly simulations (see Fig. R2).

**R02**: It would be shown and discussions in the main text along with the model's credibility in reproducing the diurnal cycle. In fact, this is also true for other locations too depends on the data availability.

**AA:** We do agree; however we do not have access to full data record (e.g. the diurnal record) from other stations.

**R02**: Please include a table in the main text that provides correlations (r2) between the model, including WRF and global reanalysis, and observations for all the stations. This will help emphasize the advantages of the regional high-resolution model.

**AA:** Please see Fig. 4 (in the revised manuscript). The values of the correlation coefficient between observation and different models (including global models) are given as the black line.

**R02**: If the model simulations are capable of generating monthly values, it would be beneficial to include these values for assessing the relative contributions to $CO_2$ variability for all four stations at a seasonal scale. This could offer insights into the seasonal characteristics of different components. Additionally, it would be advisable to incorporate the contribution of biomass burning along with other contributions.

**AA:** We analysed relative contribution with respect to seasons (please refer to additional materials Fig. AM 11-14). Please see the following lines in the manuscript about the seasonal variation in relative contribution.

L513-517: "*At the same time, the proportions of contributions to the total $CO_2$ can vary with seasons, such as winter (DJF), pre-monsoon (MAM), monsoon (JJA) and post-monsoon (SON) (Figures not shown) due to variations in atmospheric mixing and local fluxes. For instance, the reduction in $CO_2$bio component over Shadnagar and*

*Nagpur during JJA can be very likely due to increased uptake and mixing during the monsoon period.*"

Regarding the biomass burning component: please see our response to the previous comment. The biomass burning $CO_2$ contribution, on an annual scale, is negligble in all stations except Mohali (Mohali case is separtely discussed in Sect. 6.1).

L473-481: *"Both of these data have a horizontal resolution of 0.1 × 0.1 and a temporal resolution of one day. The biomass emission fluxes over the footprint region of Mohali closely follow the fire count data (see Fig. S11a). GFAS emission fluxes over the Mohali footprint region are much less than FINNv2.5 (Fig. S11a). STILT simulations using FINN (STILT-FINN) indicate some influence from biomass burning with a time-lead with the $CO_2$ observations (see Fig. S11b). However, STILT simulations using GFAS (STILT-GFAS) could not represent the $CO_2$ contributions from biomass burning (Fig. S11b). The magnitude of $CO_2$ enhancement due to the biomass burning from STILT simulations is 3 ppm, which is ~15 times less than the magnitude of the emission enhancement observed in the $CO_2$ observations (see Fig. S11b). This suggests that the emission inventories need to be improved further to accurately simulate the $CO_2$ variability due to biomass burning in the region."*

**R02**: In the conclusion section, it is important to address the limitations in replicating fine-scale $CO_2$ variability at various locations using high-resolution models. Discuss whether these limitations stem from resolution issues or the potential for flux improvements to enhance agreement. These considerations are crucial for the development of regional carbon data assimilation system that can robustly estimate sources and sinks tailored to the Indian region.

**AA:** Thank you. We have revised as follows:

L588-589: "*Our model shows reasonable skill in representing the observed $CO_2$ variability in these stations, though the model could not sufficiently capture all the observed fine-scale variations.*"

L592-599: "*Further, we explored the limitations of the STILT modelling system in representing the variability, although the model captures the intra-seasonal variabilities much better than the global models. However, the model must account for small-scale flux variations like biomass burning to represent Mohali observations. A considerable portion of these discrepancies can be minimized by improving the prior emission flux distribution at high spatial and temporal scales. Both STILT and global models did not capture the sharp reduction of $CO_2$ concentration at Nainital during August, resulting from the increased biospheric uptake by the crops over the IGP region. An improved estimate of biospheric $CO_2$ uptake and release can be achieved by utilising these $CO_2$*

*observations to optimise the fluxes through carbon data assimilation or by using flux observations from different ecosystems to calibrate the model parameters over India (e.g., Ravi et al., 2023)"*

**Minor comments:**

**R02**: Line No 130: "The performance of the WRF model simulations over India was assessed by previous studies". The intended subject of the reference in question is unclear. It could pertain to studies related to either CO2 or meteorological variables. However, it's worth noting that numerous studies exist concerning meteorology in India. To enhance clarity, please specify the subject when revising the manuscript.

**AA:** Revised as follows:

L142-145: *"The evaluation of the WRF model simulations over India shows a good agreement with observations (e.g., Hariprasad et al., 2014; Boadh et al., 2016; Sivan et al., 2021; Mathew et al., 2023). Mathew et al. (2023) demonstrated that temperature, moisture and wind simulations by WRF had a good correlation ($R^2 > 0.75$) in comparison with observations."*

**R02**: In the methodology section, when using $CO_2$ data from various instruments (PICARRO/LGR/LICOR), it would be beneficial to include information about the associated uncertainty and limitations.

**AA:** We have included the uncertainty and limitations of PICARRO and LGR in the manuscript (please see Sect. 3.1). Unfortunately, the uncertainty values of LICOR measurements at Nagpur are not available to us.

**R02**: Line No 232: It would be more appropriate to interpolate the respective vertical levels from the model outputs that closely match the altitude of the observations, rather than simply selecting the nearest level for validation against the ground-based observations.

**AA:** We have chosen the nearest vertical level for minimizing the interpolation /smoothening errors. All the observations we analysed are ground-level stations whose maximum height above ground level is 10 m. For CarbonTracker, we have used the lowest model level. In the case of EGG4 and CarboScope, the outputs are available in the pressure levels. For Mohali, Shadnagar and Nagpur, we have used the lowest available pressure level (1000 mb). In the case of Nainital, which is a mountain site situated around 800 mb height (1940 m), we used 800 mb pressure level. We have revised the manuscript as follows for clarity:

L247-250: *"To compare with the observations, simulations from the first vertical level of the CarbonTracker are used. Model simulations at 1000 mb pressure level from CarboScope and EGG4 are used to compare the observations at Mohali, Shadnagar and Nagpur. Since Nainital is a mountain site situated at ~ 800 mb height, we compared those observations with CarboScope and EGG4 products at 800 mb vertical level."*

**Figures**

[Figure]

**Figure R1:** STILT simulations with Oceanic $CO_2$ flux from CarboScope version 2022 for different staions.

[Figure]

**Figure R2:** Comparison between three hourly and hourly simulations. Diurnal cycle of 3-hourly and 1-hourly STILT simulations over Mohali during 2017.

---

## Referee Report (RR1)

Title: Potential of using $CO_2$ observations over India in regional carbon budget estimation by improving the modelling system by Thilakan et al.

The authors have made substantial efforts to address the comments from the previous review, resulting in a manuscript that shows improvement compared to the earlier version, particularly with additional discussions on aspects related to seasonality and the diurnal cycle. However, before accepting this manuscript, I still have a few concerns that should be addressed during the revision process.

In general, the mismatches between observations and the model are influenced by emissions, encompassing both biospheric and anthropogenic sources, and uncertainties stemming from transport errors. How do you discern the variations in observations-model disparities under different environmental conditions? For instance, Mohali is influenced by anthropogenic sources, while Nainital, being a high-altitude background site, is influenced by biospheric emissions. Is there a transport-related role in the observational and model mismatches observed in Nainital? Understanding whether the errors or mismatches in the model are linked to emissions or transport is crucial.

In line 370: The authors have emphasized that the sharp decline in $CO_2$ concentrations is attributed to the uptake by Rabi crops. However, in certain instances, the authors underscore the insufficient representation of biospheric fluxes, particularly related to crops. Clarification on this statement would be beneficial.

The authors suggest an insufficient representation of biospheric fluxes in the model. It raises curiosity about what alternative biospheric fluxes might address this issue, considering that the VPRM biospheric fluxes have already been incorporated into the regional models.

Another crucial aspect to consider when simulating the model for the entire year is the significant role that the Planetary Boundary Layer (PBL) plays in the variability of observations. Was this considered in the design of the model experiments, such as incorporating a proper PBL scheme?

---

## Author Response (AR2)

**Authors' Responses to the Editor and Referees:**

We are grateful to the Editor for giving us the opportunity to address the comments from Referee #2. We thank Referee #2 (**R02**) for reviewing our revised manuscript and providing helpful comments. We have addressed all these comments and incorporated the associated modifications into the manuscript.

The Editor's and R02's comments are given in regular black font, our responses are given in regular blue font, and the changes in the revised version are given in *blue italic font*.

**EC/ RC: Editor/R02 Comments**

**AR: Authors' Response**
* * *
**EC:** Dear Authors, All reviewers agree that the revised version is much improved over the original submission. However, one of the reviewers (referee #2) have raised 4 good questions that I would like you to address. I believe that all of these can be addressed with a simple sentence or two. These are all minor updates but will add to the value of the work and make this a robust study. Kindly address these comments and then the manuscript can be accepted for publication. Thank you.

**AR:** We agree. Our responses to R02's comments are listed below, indicating the modifications in the second (now-)revised manuscript.

**Response to Referee 2 comments**

**RC:**

The authors have made substantial efforts to address the comments from the previous review, resulting in a manuscript that shows improvement compared to the earlier version, particularly with additional discussions on aspects related to seasonality and the diurnal cycle. However, before accepting this manuscript, I still have a few concerns that should be addressed during the revision process.

**AR:** Thank you for carefully reviewing our manuscript and appreciating the importance of work. Please see our responses to the referee's concerns below.

**RC:** In general, the mismatches between observations and the model are influenced by emissions, encompassing both biospheric and anthropogenic sources, and uncertainties stemming from transport errors. How do you discern the variations in

observations-model disparities under different environmental conditions? For instance, Mohali is influenced by anthropogenic sources, while Nainital, being a high-altitude background site, is influenced by biospheric emissions. Is there a transport-related role in the observational and model mismatches observed in Nainital? Understanding whether the errors or mismatches in the model are linked to emissions or transport is crucial.

**AR:** Thank you for this comment. We acknowledge that model-data mismatches are mainly due to the incorrect transport and flux variations in the model (see Sect. 6.6). Decoupling model uncertainty solely due to transport and prior (input) fluxes is challenging while evaluating the model with observations, especially when both components (transport and flux variations) contribute considerably to atmospheric $CO_2$ variations. It should be noted that both Nainital and Mohali observations have significant contributions from transport, as explained in the manuscript (see Sect. 5.1), in addition to flux variability. In the Sect. 6.4, we discussed these influences by showing the contribution of different components to the total $CO_2$ concentration. By improving the transport (STILT model driven by fine–scale meteorology), we minimised modelling errors (see Sect. 5.2) compared to reanalysis products. Further, we reported significant differences (up to 8 ppm variability) in the Mohali $CO_2$ simulation related to the choice of the emission inventory in the STILT model (Sect. 6.4). We revised the manuscript to clarify this vital point:

L55-57: “*The model-observation mismatch in atmospheric $CO_2$ concentrations emerges due to the combined effect of uncertainties in the transport processes and the improper representation of $CO_2$ flux variability.*”

L442-445: “*Observations from all these four sites show strong seasonal variations in $CO_2$ concentrations (see Sect. 5.1), contributed by biospheric flux variations and transport mechanisms. Along with the seasonal variations, these observations (except Nainital) are also characterised by strong small-scale variability associated with local flux variations and mesoscale transport processes.*”

L566-569: “*... STILT simulations is particularly relevant to assess the readiness of our models to utilise these measurements in the carbon assimilation system. Though it benefits the inverse modelling community, this study is not designed to entirely decouple the uncertainties solely due to inadequate transport and improper representation of flux variations in the model.*”

L598-599: “*By improving fine–scale transport in the model, STILT simulations agree better with the observed seasonal and diurnal variations than the global reanalysis products.*”

**RC:** In line 370: The authors have emphasized that the sharp decline in $CO_2$ concentrations is attributed to the uptake by Rabi crops. However, in certain instances, the authors underscore the insufficient representation of biospheric fluxes, particularly related to crops. Clarification on this statement would be beneficial.

**AR:** We indicated that the biosphere model could not adequately capture crop uptake, resulting in an overestimation of atmospheric $CO_2$ simulations. The text has been revised for clarity as follows:

L332-334: *"The observed decline is likely due to the increased biospheric uptake by Rabi crops during this period, which may be misrepresented in the biospheric model. This is further examined in detail in Sect. 6.5."*

**RC:** The authors suggest an insufficient representation of biospheric fluxes in the model. It raises curiosity about what alternative biospheric fluxes might address this issue, considering that the VPRM biospheric fluxes have already been incorporated into the regional models.

**AR:** As the referee indicated, the VPRM model is being widely used to represent biospheric $CO_2$ fluxes across the world. The model benefits from a network of long–term eddy-covariance flux observations for parameter optimisation, which is currently lacking in the Indian region. The availability of additional $CO_2$ flux observations or the incorporation of additional satellite observations (e.g. Solar Induced Fluorescence) may improve the model's performance.

The text is revised to include these points:

L558-560: *"Note that the VPRM model used in the present study lacks parameter optimisation against eddy-covariance flux observations across India. The availability of eddy-covariance flux observations representing various biomes in India is expected to improve the model performance.*

L606-609: *"In addition to using eddy-covariance flux observations in India, utilising additional satellite observations such as Solar Induced Fluorescence in the VPRM model can likely improve the prior representation of biospheric $CO_2$ uptake and release across Indian biomes (e.g., Ravi et al., 2023). Further, an improved (inverse) estimate of fluxes can be achieved by utilising atmospheric $CO_2$ observations through carbon data assimilation."*

**RC:** Another crucial aspect to consider when simulating the model for the entire year is the significant role that the Planetary Boundary Layer (PBL) plays in the variability of observations. Was this considered in the design of the model experiments, such as incorporating a proper PBL scheme?

**AR:** Yes. In this study, the STILT model utilised the modified Richardson number to calculate the PBL height (see Sect. 2) by using meteorological fields from the WRF model. The WRF-simulated meteorological variables are compared well with observations (e.g., Mathew et al., 2024). Evaluating the performance of STILT $CO_2$ simulations with respect to the changes in PBL simulations requires additional modelling set-ups/analysis along with the availability of PBL observations, which is not addressed in this study.

The text is modified as follows:

L57-59: "*The accurate representation of the planetary boundary layer (PBL) height is also crucial for the simulation of tracer distribution in the boundary layer and its dynamics (e.g., Gerbig et al., 2008).*"

*L131-132: "These WRF meteorological simulations (temperature, moisture and wind) are compared reasonably well (*$R^2$ > 0.75*) with observations (Mathew et al., 2024).*

**References**

- Gerbig, C., Körner, S., and Lin, J. C.: Vertical mixing in atmospheric tracer transport models: error characterization and propagation, Atmospheric Chemistry and Physics, 8, 591–602, https://doi.org/10.5194/acp-8-591-2008, 2008.
- Mathew, T. A., Ravi, A., Pillai, D., Saradambal, L., Kumar, J. S., Gopalakrishnan, M. M., and Thilakan, V.: Evaluating the meteorological transport model ensemble for accounting uncertainties in carbon flux estimation over India, EGUsphere [preprint], https://doi.org/10.5194/egusphere-2023-2334, 2024.